# MicroRNA Profiles of Maternal and Neonatal Endothelial Progenitor Cells in Preeclampsia

**DOI:** 10.3390/ijms22105320

**Published:** 2021-05-18

**Authors:** Lars Brodowski, Bianca Schröder-Heurich, Sandra von Hardenberg, Katja Richter, Constantin S. von Kaisenberg, Oliver Dittrich-Breiholz, Nadia Meyer, Thilo Dörk, Frauke von Versen-Höynck

**Affiliations:** 1Gynecology Research Unit, Hannover Medical School, Carl-Neuberg-Strasse 1, D-30625 Hannover, Germany; Brodowski.Lars@mh-hannover.de (L.B.); Schroeder-Heurich.Bianca@mh-hannover.de (B.S.-H.); vonHardenberg.Sandra@mh-hannover.de (S.v.H.); Richter.Katja@mh-hannover.de (K.R.); Meyer.Nadia@mh-hannover.de (N.M.); Doerk.Thilo@mh-hannover.de (T.D.); 2Department of Obstetrics and Gynecology, Hannover Medical School, Carl-Neuberg-Strasse 1, D-30625 Hannover, Germany; vonKaisenberg.Constantin@mh-hannover.de; 3Research Core Unit Genomics, Hannover Medical School, Carl-Neuberg-Strasse 1, D-30625 Hannover, Germany; Dittrich.Oliver@mh-hannover.de

**Keywords:** preeclampsia, cardiovascular morbidity, endothelial progenitor cells, miRNA, hsa-miR-1270

## Abstract

Preeclampsia is associated with an increased cardiovascular morbidity of mother and offspring, thus contributing to a substantial burden in women and children’s health. It has been proven that endothelial progenitor cell (EPC) numbers and functional characteristics are impaired in cardiovascular disease and preeclampsia, although causative factors for the latter have remained elusive. MicroRNA (miRNA) modifications are a potential mechanism through which exposure to an altered environment translates into the development of chronic disease. In this study, we examined whether development of preeclampsia corresponds to alterations of miRNAs in maternal- and cord-blood-derived EPC. To test this end, we analyzed maternal and neonatal miRNAs via RNA sequencing from endothelial cells of preeclamptic and healthy controls in different cell culture passages. We were able to demonstrate differentially represented miRNAs in all groups. Hsa-miR-1270 showed significantly different levels in cord blood EPC from preeclampsia versus control and was negatively correlated with mRNA levels of its predicted targets *ANGPTL7* and *TFRC*. Transfection with an hsa-miR-1270 inhibitor decreased the tube formation capacity and chemotactic motility but did not change proliferation in vitro. Target predictions and gene set enrichment analyses identified alternative splicing as a significantly enriched pathway for hsa-miR-1270. The top miRNAs in three other groups were predicted to target transcriptional and developmental pathways. Here, we showed for the first time significantly different levels of miRNAs and differently represented mRNA levels of predicted target genes in EPC derived from preeclampsia. Understanding the effects of preeclampsia on the epigenetic mechanisms of EPC will be crucial and may provide initial insights for further evaluation of the benefits of therapies targeting this cell population.

## 1. Introduction

A woman’s obstetrical history is an important part of her and her offspring’s risk profile for future cardiovascular disease. Preeclampsia is a severe pregnancy-specific disorder arising in the latter half of pregnancy, manifested by new-onset hypertension with proteinuria or other organ manifestations. Preeclampsia is the second leading cause of maternal and fetal morbidity and mortality in pregnancy and has long-term, adverse health implications for both mother and offspring [1,2]. Recent studies indicate that preeclampsia is not only an independent cardiovascular risk factor for the mother, but also contributes to an increased cardiovascular risk for the offspring [2,3,4]. Substantial evidence supports endothelial cell dysfunction as being a central feature of the pathogenesis of preeclampsia [5], as well as a risk factor for adverse cardiovascular events [5,6,7].

Endothelial progenitor cells (EPC) are essential to maintain a healthy endothelium over an individual’s lifetime. EPC have been extensively studied for almost 20 years and are considered a potential marker for endothelial regeneration ability [8]. Both a decreased number and impaired function of circulating EPC have been reported in patients with cardiovascular disease [9]. The level of circulating EPC predicts the occurrence of cardiovascular events and death from cardiovascular causes and may help to identify patients at increased cardiovascular risk [6,7,10]. Decreased cell numbers and colony-forming units of maternal EPC are described as a sign of impaired endothelial repair capacity in preeclampsia [10,11]. Our own research shows that endothelial colony-forming cells (ECFC), a proliferative EPC subtype, are reduced in cord blood of infants born to women with preeclampsia [12]. Further, we described key functional differences and differential DNA methylation profiles between ECFC obtained from cord blood of preeclamptic compared to healthy pregnancies [12,13].

Micro RNAs (miRNAs) are evolutionarily conserved, noncoding RNA molecules (~22 nucleotides long, single-stranded) that regulate gene expression through base pairing with complementary sequences in their target mRNA, leading to translational repression or transcript degradation [14]. MiRNA analyses indicate that a variety of pathological tissues display miRNA expression profiles that are significantly different from normal tissues and cells [15], which may be useful for a wide range of applications in clinical diagnostics and therapy [16,17]. Furthermore, emerging evidence has uncovered miRNAs as new targets or regulators of cardiovascular medications, given the ability of miRNAs to interact with certain cardiovascular drugs [18,19]. Several reports have also proposed a role for miRNAs in pregnancy complications such as preeclampsia.

In order to capitalize on the therapeutic potential of targeting ECFC in preeclampsia, the main objective of this study was to determine the global miRNA profile of maternal-blood- and cord-blood-derived ECFC for preeclampsia and healthy pregnancies (control) using small RNA sequencing. Furthermore, we compared miRNA profiles between maternal and cord blood ECFC and examined whether cell culture passage (P) affects miRNA patterns to gain important information for future use in cell-based therapies. We additionally assessed whether known target genes of differently represented miRNAs are affected. Subsequently, control ECFC were transfected with an inhibitor of the miRNA that was most highly abased in preeclampsia versus (vs.) control and ECFC function was evaluated by tube formation and cell proliferation assays in vitro. Lastly, we explored associations between altered miRNAs and pathways affected in cardiovascular disease.

## 2. Results

### 2.1. Demographic Characteristics

Women with preeclampsia (*n* = 12) and women with healthy, uncomplicated pregnancies (*n* = 9) were enrolled for this study. Depending on the availability of maternal- and cord-blood-derived ECFC, six pregnancies were evaluated further. Demographic and clinical characteristics are summarized in Table 1. Gestational age was significantly lower in preeclampsia in both the cord blood group (*p* = 0.03) and in the group of maternal-blood-derived ECFC (*p* = 0.01). Birth weight was lower (*p* = 0.004) in the group of maternal-blood-derived ECFC in pregnancies complicated by preeclampsia. Diastolic (*p* = 0.001) and systolic blood pressures (*p* = 0.001) were significantly higher in preeclampsia group.

### 2.2. ECFC Isolation and Characterization

ECFC emerged in culture as discrete, late-outgrowth colonies displaying the characteristic cobblestone morphology. Flow cytometric results showed that ECFC in P3 and P5 were homogenous and had the typical phenotype of endothelial cells, namely CD31+, CD45−, and CD133− (Appendix A). The number of days until first appearance of colonies (~50 characteristic cells) of cord blood ECFC was significantly lower in preeclampsia, while in maternal blood ECFC colonies trended to appear later in preeclampsia compared with controls (cord blood ECFC: 7.6 ± 1.6 (control) vs. 10.8 ± 2.6 (preeclampsia); *p* = 0.04; maternal blood ECFC: 13.2 ± 4.3 (control) vs. 15.2 ± 1.7 (preeclampsia); *p* = 0.31). There were no differences in the total numbers of cord blood and maternal blood ECFC colonies (cord blood ECFC: 8.2 ± 2.3 (control) vs. 10.2 ± 6.8 (preeclampsia); *p* = 0.5; maternal blood ECFC: 1.5 ± 0.8 (control) vs. 1.8 ± 1.3 (preeclampsia); *p* = 0.69) (Appendix A).

### 2.3. MiRNA Profiles in the Comparison Groups

Our study had three objectives. We first determined whether pregnancies complicated by preeclampsia have different miRNA profiles compared to healthy pregnancies. Furthermore, we analyzed if there are differences in the miRNA profiles between maternal and cord-blood-derived ECFC, as well as between P3 and P5. Venn diagrams depict the number of common and different miRNA expression changes in the different sets of samples, with two miRNAs (hsa-miR-2467-5p and hsa-miR-4421) among the three different sets of comparisons being consistently found to be altered (Appendix A).

#### 2.3.1. MiRNA Differences between Preeclampsia and Healthy Pregnancies

We first explored ECFC by miRNA sequencing for differences in miRNA expression in the preeclampsia vs. control (healthy pregnancy) groups. This pairwise analysis was conducted independently for maternal blood and cord blood, as well as for P3 and P5. The levels of 17 miRNAs derived from cord blood at P3 were notably different between preeclampsia and control (*p* < 0.05; Figure 1). Four of those miRNAs additionally fulfilled the applied fold change criterion (>2-fold), although none of them passed the Benjamini–Hochberg correction at *p* < 0.05 (Figure 1a). Comparing cord-blood-derived miRNA levels at P5, 47 miRNAs showed a notably different expression in the preeclampsia group (*p* < 0.05), but again none of these passed the Benjamini–Hochberg correction (Figure 1b). Using the same analysis for maternal blood samples, 39 miRNAs in P3 (Figure 1c) and 17 miRNAs in P5 (Figure 1d) were notably regulated. Again, none of these miRNAs passed the Benjamini–Hochberg correction. Heat maps of significantly regulated miRNAs in ECFC of controls compared to preeclampsia are shown in Appendix A. The most strongly regulated miRNAs passing the moderated *t*-test (*p* < 0.05, fold change >2) are shown in Table 2.

#### 2.3.2. MiRNA Differences between Maternal and Cord Blood

Next, we compared miRNA profiles between maternal blood and cord blood ECFC in P3 and P5 cells, with each divided into preeclampsia patients and controls (healthy pregnancies). This analysis resulted in 57 notably regulated miRNAs in P3 and 108 notably regulated miRNAs in P5 (*p* < 0.05) in ECFC derived from controls, with 4 of these miRNAs passing the Benjamini–Hochberg correction (Figure 2a,b). Comparison of miRNA profiles of maternal and cord blood ECFC derived from preeclampsia resulted in 78 notably regulated miRNAs in P3 and 163 notably regulated miRNAs in P5 (*p* < 0.05). Six miRNAs in P3 and 27 miRNAs in P5 passed Benjamini–Hochberg correction, respectively (Figure 2c,d). Heat maps of significantly regulated miRNAs in ECFC of cord blood compared to maternal blood are shown in Appendix A.

The most strongly regulated miRNAs that passed Benjamini–Hochberg correction can be found in Appendix A.

#### 2.3.3. MiRNA Differences between Cell Culture Passages 3 and 5

Further, we asked whether differential miRNA levels remained stable and whether additional differences appeared with increasing cell passage. The comparison of the miRNA profiles of cord-blood-derived ECFC between P3 and P5 resulted in 19 notably regulated miRNAs derived from controls (healthy pregnancies) and 13 miRNAs in preeclampsia (*p* < 0.05). None of these miRNAs passed Benjamini–Hochberg correction (Figure 3a,b). Comparison of miRNA profiles of maternal ECFC of P3 and P5 resulted in 10 (controls) and 5 (preeclampsia) notably regulated miRNAs. None of these miRNAs passed Benjamini–Hochberg correction (Figure 3c,d). Heat maps of significantly regulated miRNAs in ECFC of P3 compared to P5 are shown in Appendix A.

A list of miRNAs with more than two-fold changes and significantly different levels (*p* < 0.05) between P3 and P5 is provided in Table 3. Interestingly, only one miRNA (hsa-miR-3911) was shared between preeclampsia and healthy controls (control: fold change: –2.16; *p* = 0.007; preeclampsia: fold change: −21.26; *p* = 0.02), while 18 passage-associated miRNAs appeared as specific for the respective condition.

### 2.4. Quantitative Real-Time PCR Validation and Putative Target Gene Levels

The profiles of the miRNAs with the most significant differences (lowest *p*-value) between preeclampsia and the control (healthy pregnancies) group, namely hsa-miRNA-1270 (P3: *p* = 7.84 × 10^−3^, fold change: −2.63) and hsa-miR-2467-5p (P5: *p* = 5.21 × 10^−3^, fold change: 3.19) in cord-blood-derived ECFC, and hsa-miR-214-5p (P3: *p* = 2.38 × 10^−3^, fold change: −10.66) and hsa-miR-3177-5p (P5: *p* = 1.44 × 10^−3^, fold change: 2.19) in maternal-blood-derived ECFC, were further validated. QRT-PCR findings confirmed the results obtained by RNA sequencing for all four tested miRNAs (Figure 4a–d).

Several genes have been reported as putative targets for the aforementioned miRNAs and are shown in Appendix A. Since we did not find much difference in miRNA profiles between P3 and P5, we chose to study only P3 cells for deeper validation, as they most closely resemble the phenotype of freshly isolated cells, and therefore are more comparable to cells in vivo than primary cells in P5. We selected *NOSTRIN* among the target genes of hsa-miR-214-5p because it has already been shown that *NOSTRIN* is increased in placental tissues of preeclamptic patients [20]. Since we could not detect any differences in the analysis of *NOSTRIN* hsa-miR-214-5p between preeclampsia vs. control (*NOSTRIN*: preeclampsia vs. control: 1.8 × 10^−6^ vs. 1.0 × 10^−6^; *p* = 0.07) (Figure 5a), we analyzed target genes for hsa-miR-1270 (*ANGPTL7* and *TFRC*). *TFRC* was chosen because it gave the highest score for hsa-miR-1270 on TargetScan and *ANGPTL7* was selected because it is related to angiogenesis, and thus has an important link to preeclampsia. We selectively explored whether the levels of *ANGPTL7* or *TFRC* correlated with hsa-miR-1270 in cord blood ECFC. We found significantly higher *ANGPTL7* and *TFRC* mRNA levels in cord blood ECFC from preeclampsia compared to control (*ANGPTL7*: preeclampsia vs. control: 1.1 × 10^−5^ vs. 5.7 × 10^−6^; *p* = 0.02, Figure 5b; *TFRC*: preeclampsia vs. control: 1.9 × 10^−7^ vs. 3.8 × 10^−8^; *p* = 0.009, Figure 5c).

### 2.5. Pathway Enrichment Analysis

To gain more insight into possible biological functions regulated by the miRNA that displayed the most differential abundance between preeclamptic and healthy pregnancies, we investigated the whole set of predicted targets using pathway enrichment analyses. First, we determined the set of predicted targets for the miRNA using TargetScan v.7.2 and miRDB, respectively. To reduce the number of false positives, we then subjected only those miRNA targets to further pathway analyses that were shared by both prediction tools (Figure 6 and Appendix A).

TargetScan predicted 4322 transcripts with sites for hsa-miR-1270, while miRDB predicted 442 targets, of which 402 were in common with TargetScan (Appendix A). This common set of targets was significantly enriched in genes with the UniProt term alternative splicing (KW-0025, FDR = 0.00013) (Appendix A).

We performed similar analyses with the top-ranking miRNAs from three other comparison groups of preeclampsia vs. control samples (Table 2). The hsa-miR-2467-5p had 485 predicted targets shared by TargetScan and miRDB that were notably enriched in phosphoproteins (KW-597, FDR = 0.007) and the GO term regulation of cell development (GO:0060284, FDR = 0.04). The hsa-miR-214-5p had 60 predicted targets shared by both TargetScan and miRDB, but these were enriched in transcription factors (GO:0044798, FDR = 0.007). Finally, the hsa-miR-3177-5p had 517 predicted targets shared by TargetScan and miRDB, which were markedly enriched for RNA transcription (KW-0804, FDR = 5 × 10^−7^). Additional significantly enriched pathways for hsa-miR-3177-5p included cell adhesion, alternative splicing, and Wnt signaling (Figure 6, Appendix A).

### 2.6. Hsa-miR-1270 Dependent Tube Formation, Proliferation and Chemotactic Motility

Endothelial cell function was assessed by a tube formation assay (Figure 7), which assessed the capability of endothelial cells to form capillary-like structures. ECFC were transfected with a hsa-miR-1270 inhibitor to mimic the observed down-regulation in ECFC. The specific hsa-miR-1270 inhibitor is a small specific RNA molecule that binds and inhibits hsa-miR-1270. Furthermore, a positive control for transfection efficiency was also included, which leads to an increase in *HMGA2* in the qRT-PCR (Appendix A). A negative control and a vehicle control (transfection reagent alone) were used to identify non-specific off-target effects. We examined whether hsa-miR-1270, which exhibited the greatest fold change in preeclampsia vs. control in cord-blood-derived cells, affects endothelial cell function. All ECFC were able to form tubes. Hsa-miR-1270 inhibition significantly reduced the total tube lengths between the groups (untreated control: 1.64 × 10^7^ µm ± 1.8 × 10^6^ µm; negative control: 1.63 × 10^7^ µm ± 1.3 × 10^6^ µm; hsa-miR-1270 inhibitor: 1.39 × 10^7^ µm ± 2.4 × 10^6^ µm; untreated control vs. miR-1270 inhibitor: *p* = 0.028; negative control vs. hsa-miR-1270 inhibitor: *p* = 0.025) (Figure 7b). Transfection with hsa-miR-1270 inhibitor did not significantly change the number of formed loops as compared with the negative control (untreated control: 51.0 ± 10.8; negative control: 55.0 ± 11.6; hsa-miR-1270 inhibitor: 47.9 ± 11.5; untreated control vs. hsa-miR-1270 inhibitor: *p* = 0.49; negative control vs. hsa-miR-1270 inhibitor: *p* = 0.24) (Figure 7a). The total branching points tended to decrease after transfection as compared with control but the difference was not significant (untreated control: 50.6 ± 13.9; negative control: 53.6 ± 8.3; hsa-miR-1270 inhibitor: 45.6 ± 11.3; untreated control vs. hsa-miR-1270 inhibitor: *p* = 0.44; negative control vs. hsa-miR-1270 inhibitor: *p* = 0.13) (Figure 7c).

Hsa-miR-1270 inhibition did not significantly change proliferation of ECFC in vitro as compared with control ECFC (population doubling time: untreated: 18.35 h; negative control: 17.92 h; hsa-miR-1270 inhibitor: 17.81 h; *p* = 0.32) (Figure 8a), however the chemotactic motility levels of miR-1270-inhibited ECFC were impaired compared to negative or untreated control (Figure 8b) (untreated control: 84.1 cells/area; negative control: 64 cells/area; hsa-miR-1270 inhibitor: 48.6 cells/area; untreated control vs. miR-1270 inhibitor: *p* = 0.003; negative control vs. hsa-miR-1270 inhibitor: *p* = 0.25).

## 3. Discussion

To our knowledge, this is the first study demonstrating that maternal-blood- and cord-blood-derived endothelial progenitor cells from preeclamptic pregnancies display an aberrant miRNA profile compared to healthy pregnancies. In this work, we focused on hsa-miR-1270, as it was one of the most statistically different miRNAs, with a high fold change in cord-blood-derived EFCF from preeclamptic vs. healthy mothers.

MiRNA studies of human-pregnancy-related tissues and cells have started to reveal important insights into pathogenesis, however whether aberrant miRNA patterns are involved in the functional changes observed in ECFC from preeclamptic pregnancies and if these lead to an increased cardiovascular risk are factors that have not been investigated. Previous studies on plasma and placental miRNAs have indicated the involvement of miRNAs in the pathophysiology of preeclampsia [21,22]. Although some reports [23,24,25,26,27] have already revealed the potential usefulness of measuring certain miRNAs for predicting the occurrence of preeclampsia, there have been few overlaps among those studies, and more investigation is required in order to clarify the exact role of preeclampsia-related miRNAs. This also applies to our study, in which two miRNAs (miR-139-5p, miR-148a) were detected that were reported to be down-regulated in other studies in preeclampsia but investigated in different tissues (placenta, decidua-derived mesenchymal stem cells) [28,29].

In our study, 17 miRNAs in P3 and 47 miRNAs in P5 showed a significantly different representation for cord-blood-derived preeclamptic ECFC when compared to control. The smallest *p*-value here was obtained for hsa-miR-1270, although this difference did not survive correction for multiple testing. In maternal ECFC, we found 39 differentially represented miRNAs in P3 and 17 in P5. Considering the therapeutic potential of ECFC, an essential aspect of our research was also the comparative analysis of two cell culture passages of the progenitor cells. P3 reflects the first passage after cell isolation that provides the required quantity and quality of ECFC for this and other analyses. Since primary cells are subject to aging, which is increasingly observed in higher passages, e.g., 7–8 onwards, we deliberately chose P5 as a comparison to avoid possible developmental changes. Here, we observed passage-associated changes in miRNA levels that were different in preeclampsia-derived ECFC compared with control ECFC, with only hsa-miR-3911 shared between both conditions. These observations suggest that there may be disease-associated changes affecting the propagation of preeclamptic ECFC in cell culture. The information gained is important, as it suggests that the priming of ECFC persists over cell culture passages and developmental effects play a minor role at least up to P5. Among the three different set of comparisons, hsa-miR-2467-5p and hsa-miR-4421 were consistently found to be altered among all of these comparisons.

Hsa-miR-1270 showed the smallest *p*-value in the preeclampsia group compared to the control. These results were further validated in vitro and putative target genes were investigated. The reduced level of miRNA hsa-miR-1270 observed by RNA sequencing in cord blood ECFC was confirmed by qRT-PCR. Interestingly, some predicted target gene products are also associated with preeclampsia, including the transferrin receptor, TFRC. TFRC is necessary for cellular iron uptake and has previously been noted to have markedly reduced expression in placentae from preeclamptic pregnancies compared to those from healthy pregnancies [30]. The level of *TFRC* mRNA was found to be significantly different in preeclampsia samples in our study, suggesting that *TFRC* might be regulated under steady-state conditions by hsa-miR-1270 in ECFC. A similar although less pronounced effect was observed for *ANGPTL7*.

According to our knowledge, only two studies have so far investigated the functional role of hsa-miR-1270, with conflicting results. While down-regulation of hsa-miR-1270 in papillary thyroid cancer cell lines TPCI and K1 suppressed cell proliferation, migration, and in vivo transplantation [31], up-regulation of hsa-miR-1270 suppressed human glioblastoma cancer cell proliferation, migration, and tumorigenesis [32]. Limited data exist on the effects of siRNA-mediated inhibition of *TFRC* and *ANGPTL7*. Increase of VEGF was observed in *TFRC* knockdown MDA-MB-231 cells when compared to the parental cells. In human primary human trabecular network cells, down-regulation of *ANGPTL7* leads to significantly lower levels of extracellular matrix protein MMP1 but increases in fibronectin, collagen type VA1, myocilin, and versican [33]. As there are a magnitude of possible mechanisms and targets (see discussion of pathway enrichment analysis below), further investigation into the potential role of hsa-miR-1270 is required.

In a more global approach and through gene set enrichment analysis, we evaluated whether the sets of predicted targets of hsa-miR-1270 (and top-ranking miRNAs for the three other comparisons) mark specific pathways and protein families, which may be implicated in preeclampsia. The targets for hsa-miR-1270 and hsa-miR-3177-5p were enriched for alternative splicing, indicating that they might regulate gene expression. Alternative splicing of certain proteins has been implicated previously in the pathophysiology of preeclampsia, with the most notable example being s-Flt (for a review see Palmer et al., 2017 [34]). However, there is also evidence of a more global dysregulation of alternative splicing in preeclampsia [35]. Consistent with a role of the differentially represented miRNAs in gene expression, further enrichments were observed for RNA transcription and developmental pathways.

Our analyses were limited to only the top-ranking miRNAs, because none of the miRNAs passed the strict Benjamini–Hochberg correction. We extended our observations by a mechanistic approach and demonstrated that preeclampsia-specific miRNAs, e.g., hsa-miR-1270, affect endothelial progenitor cell angiogenic function in vitro. We acknowledge that the in vitro transfection used in our study differs from the situation in vivo. However, using miRNA inhibitors to analyze endogenous miRNA function is widely accepted as a tool to study the effects of different miRNA levels [36]. Although we observed a significant effect of hsa-miR-1270 inhibition on ECFC tubule length and chemotactic motility, no significant difference was noted in cell doubling. This effect seems biologically plausible. Compared to cell proliferation, tubule formation is a biologically more sophisticated process that requires a coordinated interaction between cells. Limitations of this complicated process may be of consequence much earlier. While hsa-miR-1270 seems to affect one part of the angiogenic process, further work will be needed to elucidate how the dysregulation of other identified miRNAs impact functional processes of EPC. Furthermore, we analyzed just one miRNA for functional differences in this study. A large-scale analysis such as luciferase binding assays in a larger study population would be useful to investigate miRNA-mediated regulation of target genes more specifically in preeclampsia. Our sample was limited to six cases per group, which is due to the well-described low isolation efficiency of ECFC. However, other publications on circulating [37] and ECFC-derived miRNAs [38] in pregnant populations provided new and interesting results, even with lower sample numbers. In addition, the comparison of mother–child pairs in both study populations would be desirable, as this would provide more stable conditions and avoid possible bias. While in our control group we were able to isolate ECFC from three mother–child pairs, none were available in the preeclampsia group. Regardless, the comparison of maternal and cord blood ECFC is a good starting point that should be expanded on in the future.

Nowadays, it is assumed that preeclampsia is essentially a disease of the vascular endothelium, which in turn contributes to an increased cardiovascular risk in later life. Discovered in the late 1990s, EPCs attracted clinical and basic research as key regulators of vascular homeostasis in health and disease, with their ability for neovasculogenesis, angiogenesis, and endothelial repair [8,39]. Since the first discovery of miRNAs, their involvement in different aspects of vascular disease has emerged as an important research field. MiRNAs have been found to be critical modulators of endothelial homeostasis, while dysregulation of miRNAs has been linked to endothelial dysfunction and the development and progression of vascular disease. Furthermore, insufficient angiogenesis is characterized by ischemic heart disease, peripheral vascular disease, and preeclampsia [40,41]. The relevance of miRNAs in vascular neovascularization has been demonstrated by different approaches of knockdown enzymes involved in the biogenesis of miRNAs [42,43,44]. As miRNAs are known to be critical in fine-tuning and in maintaining the physiological balance of the vascular endothelium, they are targets for miRNA-based therapies via reprogramming of endothelial cells. Although the effects of these potential therapeutic agents for endothelial function remain to be properly assessed, it is of crucial importance to first identify the potentially pathogenic miRNAs, such as in this pilot study.

In summary, we identified differentially expressed miRNA profiles that allowed us to predict target genes and pathways in preeclampsia that may be involved in the pathogenesis of the disease. Considering the small patient cohort, the results can serve as a basis for follow-up studies with a larger sample and a more limited selection of candidates to avoid the disadvantage of multiple testing. Epigenetic modifications are one of the potential mechanisms, including aberrant miRNA expression, through which the exposure to an altered in utero environment translates into the development of chronic disease. Understanding the impact of preeclampsia on epigenetic mechanisms and response of ECFC to therapeutic strategies will be critical in evaluating the utility of therapies designed to target this cell population.

## 4. Materials and Methods

### 4.1. Participants

We recruited subjects under Ethical Approval Number 3254 (Hannover Medical School) from the Division of Obstetrics. After informed consent, pre-delivery maternal blood and cord blood were collected and ECFC of 12 preeclamptic pregnancies (*n* = 6 from maternal; *n* = 6 from cord blood with no maternal-cord blood pairs) and 9 healthy pregnancies (*n* = 6 from maternal; *n* = 6 from cord blood with 3 maternal-cord blood pairs) were isolated. Healthy participants had no significant maternal or neonatal disease. All were singleton pregnancies matched for gestational age at delivery, maternal BMI, and maternal age. Preeclampsia was defined as new-onset hypertension ≥140/90 mmHg on two or more occasions after 20 weeks gestation accompanied by proteinuria or other symptoms of organ dysfunction, e.g., liver or acute kidney dysfunction, neurological symptoms, hemolysis, thrombocytopenia, or fetal growth restriction [45]. Resting blood pressure levels were taken by a trained nurse using the oscillometric method and the appropriate cuff size.

### 4.2. ECFC Isolation and Characterization

Primary human ECFC were isolated, characterized, and cultivated as previously described by others and our own studies [12,46,47]. Briefly, maternal peripheral (PBMC) or venous cord blood mononuclear cells (CBMC) were isolated by density gradient centrifugation and PBMC or CBMC were plated (5 × 10^7^ cells/well) onto collagen-coated 6-well plates (BD Bioscience, Heidelberg, Germany) containing endothelial growth medium 2 (EGM-2, Lonza, Basel, Switzerland), supplemented with the supplier’s recommended concentrations of growth factors (10% FBS and 1% penicillin–streptomycin). After 10–21 days of cultivation, ECFC appeared as adherent single layers of cobblestone-shaped, late-outgrowth cells that formed colonies (>50 cells). Flow cytometric analyses to confirm the ECFC phenotype were performed using surface markers CD31, CD133, and CD45, as well as appropriate isotype controls (Appendix A).

### 4.3. Small RNA Sequencing

The global miRNA profiles of maternal-blood- and cord-blood-derived ECFC from six preeclamptic pregnancies and six uncomplicated, healthy pregnancies were determined by small RNA sequencing. For miRNA isolation, cultured ECFC were lysed on ice and RNA was extracted according to the recommended protocol variant for total RNA including small species (mirVANA Kit, Ambion, MA, USA). Library preparation for small RNA sequencing was performed by use of the NEBNext^®^ Multiplex Small RNA Library Prep Kit for Illumina^®^ 96 rxns, Index Primers 1–48 (E7560S, New England Biolabs, Frankfurt am Main, Germany).

Forty-eight libraries (reflecting 48 biological samples) were specifically barcoded, pooled, denatured, and diluted. Two denatured pools, each containing 24 libraries, were run on an Illumina NextSeq 550 sequencer, using two high-output flow cells for 75 bp single reads (San Diego, IL, USA) and yielding 15 million reads on average per analyzed sample (median: 13.5 million). Raw data processing and quality control were conducted with bcl2fastq Conversion Software, Trim Galore, and FastQC, while mapping, quantification, normalization, and differential expression analyses were performed with StrandNGS v3.1 (Avadis, Bangalore, Karnataka, India). Levels of 2104 miRNAs were considered in the analysis based on their read counts. In order to eliminate undetectable or poorly reliable miRNA signatures, data were positively filtered for miRNAs, showing at least two normalized read counts in all samples of at least one out of the eight analyzed groups, namely maternal- or cord-blood-derived, cell passage 3 or 5, and control or preeclampsia. This first expression filter yielded 703 miRNAs that were deemed expressed, which served as a starting point for subsequent analyses. The Shapiro–Wilk test was used to investigate whether resulting expression data were distributed normally. For miRNAs with *p*-values greater than the chosen alpha level (0.01), we accepted the null hypothesis that the data came from a normally distributed population. These miRNAs were further analyzed using the moderated *t*-test (*p* < 0.05). In addition, *p*-values were corrected using the Benjamini–Hochberg procedure to account for multiple testing and to estimate the false discovery rate (FDR). Normalized log-2-transformed count data were subsequently baseline transformed to the median. These data were imported into Omics Explorer v3.6 (Qlucore, Lund, Sweden) for final heatmap visualization. More detailed information on RNA sequencing can be found in Appendix A.

MiRNA profiles of ECFC from preeclamptic and healthy pregnancies were compared. Separate analyses were performed in ECFC derived from cord blood (offspring group, *n* = 6) and ECFC derived from maternal blood (maternal group, *n* = 6). All analyses were performed at two cell culture passages, P3 and P5. In a second approach, we compared ECFC isolated from maternal blood with ECFC isolated from cord blood. Separate analyses were performed in ECFC derived from preeclampsia and healthy pregnancies (the control group). Again, these analyses were performed at two cell culture passages, P3 and P5. Thirdly, we examined whether cell culture passage affects miRNA patterns overall. P3 corresponded to the earliest time when sufficient cell numbers were available after primary cell isolation, while P5 was selected to fall within a common window where ECFC are used for functional studies [37,41,42]. Separate analyses were performed in ECFC derived from cord blood (offspring group) and ECFC derived from maternal blood (maternal group). In this approach, the analyses were stratified by disease status. For further validation and target gene expression analysis of maternal and cord blood miRNAs, we first focused on the biologically relevant effect sizes by filtering for miRNAs with log-fold changes ≥2, followed by the most significant hit (lowest *p* value), independent of its effect size, which is a common procedure in case–control association studies. For the final selection step, the highest fold change was considered.

### 4.4. qRT-PCR for miRNA Validation and Putative Target Gene Expression

To evaluate the miRNA level, an miScript Primer Assay (Qiagen, Leipzig, Germany, Cat. No.: MS00014392 (hsa-miR-1270)) was used according to the manufacturer’s recommendations to convert RNA for to cDNA synthesis. The qRT-PCR was conducted with miScript II Master Mix (miScript II, SYBR Green Assay, Qiagen, Leipzig, Germany, 218160) on a Corbett Rotor Gene (Corbett Life Science, Sydney, Australia).

Putative target genes of differently represented miRNAs were predicted via TargetScan (http://www.targetscan.org/vert_72/, accessed on 24 July 2019) [48] and further analyzed for mRNA level via qRT-PCR. For normalization, 18S rRNA (*RNA18S1*) served as the housekeeping gene.

Primer sequences for determination of mRNA and miRNA levels are described in Appendix A. For qRT-PCR analysis of miRNA, the miScript Primer Assays with miRNA-specific forward primer (Appendix A) and the miScript SYBR Green PCR Kit, containing the miScript Universal Primer (reverse primer) and QuantiTect SYBR Green PCR Master Mix, were used. Relative quantification of gene expression was calculated by standard ΔCt method using the expression of 18S rRNA (target genes) or *RNU6* (miRNA) as the reference. For each treatment and qRT-PCR analysis of miRNA and mRNA, runs were performed in triplicate. A total of three RT-PCR runs were performed for each patient. Relative levels in six ECFC from the respective groups were compared.

### 4.5. Transfection of ECFC with miRNA Inhibitor

Down-regulation of hsa-miR-1270 was observed in our study. To analyze this effect in vitro, we used a miRNA inhibitor, a small RNA molecule that inhibits miRNA by binding. At 70–85% confluence, ECFC were transfected with either *mir*Vana™ miR-1270 inhibitor (ID: MH13389), *mir*Vana™ miRNA inhibitor *let-7c* positive control (#4464080), *mir*Vana™ miRNA inhibitor negative control #1 (4464076) (Thermo Fisher Scientific, Waltham, MA, USA), or vehicle control (transfection reagent alone: Lipofectamine RNAiMAX, Invitrogen, Waltham, MA, USA) according to the manufacturer’s protocol. The *mir*Vana™ miRNA inhibitor *let-7c* positive control was used to determine transfection efficiency. After transfection into cells, the *let-7c* miRNA inhibitor blocks endogenous *let-7c* miRNA, leading to elevated *HMGA2* mRNA levels, which are analyzed by qRT-PCR. The *mir*Vana™ miRNA inhibitor *let-7c* positive control was transfected in parallel with the miR-inhibitor or *mir*Vana™ miRNA inhibitor negative control #1 (Appendix A). For transfection, 9 µL Lipofectamine RNAiMAX reagent was diluted in 150 µL Opti-MEM Medium (Life Technologies, Waltham, MA, USA) and 30 nM miRNA inhibitor or related controls, which were diluted in 150 µL Opti-MEM medium. Cells were incubated for 24 h at 37 °C and 5% CO_2_ before they were used for functional assays. Transfection efficiency was determined via qRT-PCR by expression of *HMGA2* mRNA (positive control). Then, 24 h after transfection, ECFC were analyzed for tube formation capacity and proliferation ability.

### 4.6. Tube Formation Assay

An in vitro angiogenesis assay was used to test the capacity of ECFC to form capillary tubule-like networks. Here, 1.7 × 10^4^ cells/well of hsa-miR-1270-inhibitor-transfected ECFC were seeded in a 96-well plate containing treatment medium and growth-factor-reduced Matrigel matrix (BD Biosciences, Bedford, MA, USA) and transferred onto a LEICA DMI 6000 B microscope equipped with Incubator BL (Leica, Wetzlar, Germany) for heating and 5% CO_2_ supply. Images were taken by using phase contrast optics with a 2.5× objective after 12 h. The total tubule length in each visual field, quantity of whole closed loops, and number of interconnections between the tubules (branching points) were measured using Image J software 1.52q (National Institutes of Health, Bethesda, MA, USA). The total tubule length describes the sum of all tubule fibers of the entire image section. The quantity of whole closed loops is the number of all closed tubular loops in the image section. Branching points were defined as points of interaction from which at least three tubules originated.

### 4.7. Cell Proliferation Assay

To determine the proliferative capacity of ECFC after inhibition of hsa-miR-1270, 2.5 × 10^4^ cells were seeded per well of 24-well culture plates in EGM supplemented with 5% (*v*/*v*) FBS and 1% penicillin–streptomycin. After 12, 24, and 48 h, the cell number and population doubling time were determined by trypan blue staining. ECFC collected after 24 h of transfection and a corresponding untreated and negative control were run in tandem.

### 4.8. Chemotaxis Assay

Chemotaxis assay was performed by using transwell inserts with a translucent microporous membrane (ThinCerts, Greiner, Kremsmünster, Austria), as recently described [49]. After transfection with hsa-miR-1270 inhibitor or related controls, cells were serum-deprived for 24 h with growth medium supplemented with 2.5% FCS. After 24 h, cells were trypsinized and 7 × 10^4^ cells were seeded into the apical side of the insert in serum-free medium. As a chemoattractant, medium supplemented with 10% FCS was applicated in the basolateral side of the insert and cells were placed in an incubator for 4 h. Non-migrated cells were removed with a cotton bud from the apical side of the insert, while migrated cells on the insert membrane were fixed in 3% PFA, 2% sucrose in PBS for 10 min, washed two times with PBS, and counterstained with 4′,6-diamidin-2-phenylindol ((DAPI) (Invitrogen, Waltham, MA, USA)). After mounting in antifade fluorescence mounting medium (ProLongGold, Thermo Fisher Scientific, Waltham, MA, USA), five pictures per condition were randomly taken and DAPI-stained cells per area were counted to analyze chemotactic migration. 

### 4.9. Statistics and Bioinformatics Analyses

Data distribution was compared between groups using the Shapiro–Wilk test. Differences in the mean and median levels of continuous data were tested using Mann–Whitney U test or Wilcoxon two-sample test as appropriate. Categorical variables were compared by Fisher’s exact test. All *p*-values were two-sided, and *p*-values less than 0.05 were considered statistically significant. Where multiple testing had to be considered, *p*-values less than 0.05 were considered noteworthy before FDR correction and considered statistically significant if they remained less than 0.05 after FDR correction.

We compared predicted targets for miRNAs with different levels of abundance in preeclampsia vs. healthy pregnancies at P3. Target predictions were accomplished using the Target Scan 7.2 database (http://www.targetscan.org/vert_72) [48] and the miRDB database (http://mirdb.org/) [50]. Target sets were then investigated using the database of the Gene Ontology (GO) Consortium (www.geneontology.org/) [51] for an enrichment of biological pathways as defined in the Kyoto Encyclopedia of Genes and Genomes (KEGG) database (www.genome.jp/kegg/) [52], and for predominant protein domains according to the PFAM database (https://pfam.xfam.org/) [53]. These resources are publicly accessible and are implemented in STRING (www.string-db.org/) [54]. All databases were accessed on 24 July 2019. Biological processes, pathways, or protein domains were considered significantly enriched after correction for multiple testing if the FDR was <0.05.

## 5. Conclusions

The data obtained are novel both in terms of the cells studied and the disease pathogenesis investigated. Furthermore, this study lays the foundation for future work investigating whether miRNA differences contribute to cardiovascular risk in women and offspring from pregnancies complicated by preeclampsia. Considering the potential of EPC-based cytotherapies, further studies on their miRNA modifications by complicated pregnancies are warranted to develop miRNA-based therapeutics targeting cardiovascular complications. An epigenetically modified endothelial precursor during embryo- and fetogenesis may affect both normal morphogenesis and post-natal physiological adult progenitor-mediated vascular repair. Therefore, the miRNA status of ECFC may be key to understanding the pathogenesis of cardiovascular changes after preeclampsia, as well as the ultimate long-term health of mother and offspring.

## Figures and Tables

**Figure 1 ijms-22-05320-f001:**
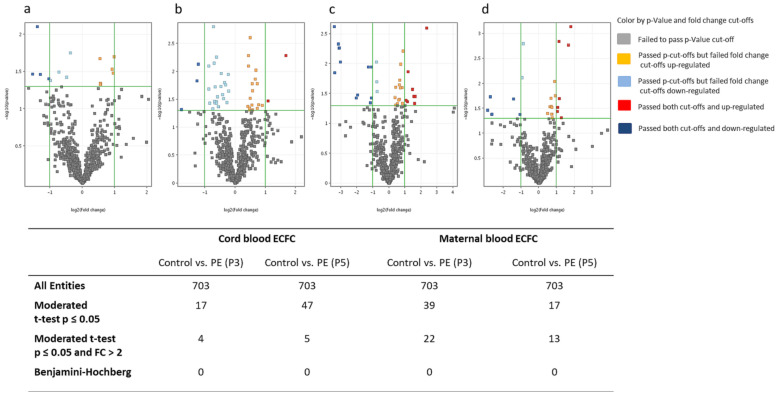
Significantly regulated miRNAs in ECFC of controls compared to preeclampsia subjects. Volcano plots show the fold changes of miRNAs on the x-axis and the statistical significance on the y-axis in relation to the reference group. Control: healthy pregnancy; PE: preeclamptic pregnancy; P3: passage 3; P5: passage 5; *n* = 6 in each group. (**a**) Control vs. PE in P3 in cord-blood-derived ECFC. (**b**) Control vs. PE in P5 in cord-blood-derived ECFC. (**c**) Control vs. PE in P3 in maternal-blood-derived ECFC. (**d**) Control vs. PE in P5 in maternal-blood-derived ECFC.

**Figure 2 ijms-22-05320-f002:**
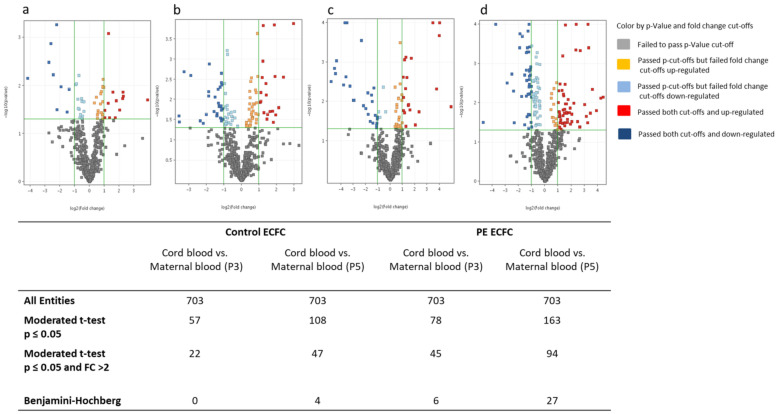
Significantly regulated miRNAs in ECFC of cord blood vs. maternal blood. Volcano plot shows the fold changes of miRNAs on the x-axis and the statistical significance on the y-axis in relation to the reference group. Control: healthy pregnancy; PE: preeclamptic pregnancy; P3: passage 3; P5: passage 5; *n* = 6 in each group. (**a**) Control cord-blood-derived ECFC vs. maternal-blood-derived ECFC in P3. (**b**) Control cord-blood-derived ECFC vs. maternal-blood-derived ECFC in P5. (**c**) PE cord-blood-derived ECFC vs. maternal-blood-derived ECFC in P3. (**d**) PE cord-blood-derived ECFC vs. maternal-blood-derived ECFC in P5.

**Figure 3 ijms-22-05320-f003:**
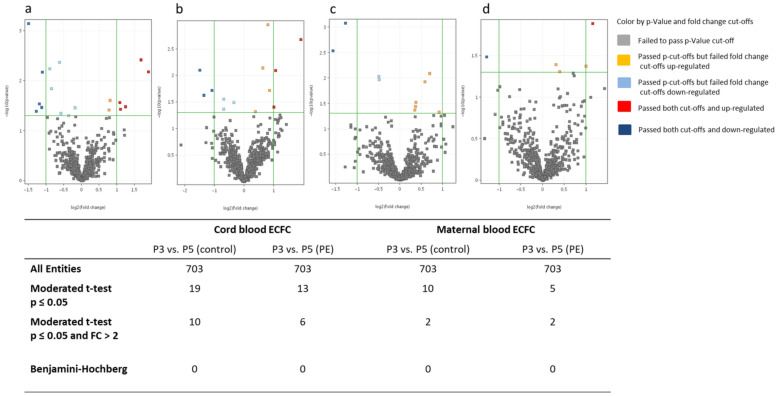
Significantly regulated miRNAs in ECFC of P3 vs. P5. Volcano plot shows the fold changes of miRNAs on the x-axis and the statistical significance on the y-axis in relation to the reference group. Control: healthy pregnancy; PE: preeclamptic pregnancy; P3: passage 3; P5: passage 5; *n* = 6 in each group. (**a**) P3 vs. P5 in control cord blood ECFC. (**b**) P3 vs. P5 in PE cord blood ECFC. (**c**) P3 vs. P5 in control maternal blood ECFC. (**d**) P3 vs. P5 in PE maternal blood ECFC.

**Figure 4 ijms-22-05320-f004:**
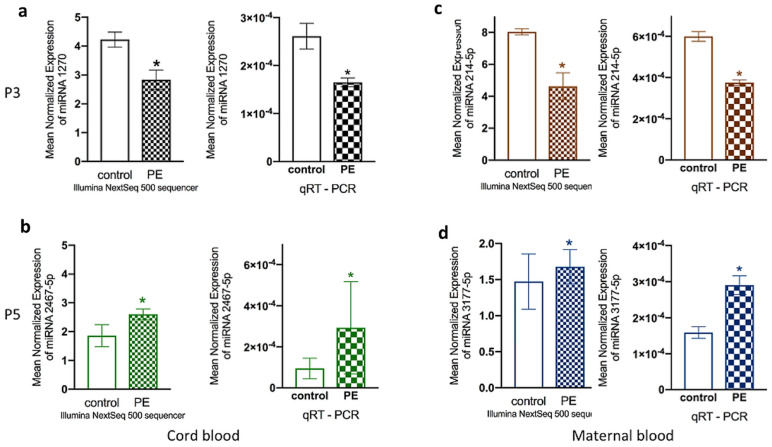
RNA sequencing and qRT-PCR results for hsa-miR-1270 (cord blood; P3) (**a**), hsa-miR-2467-5p (cord blood; P5) (**b**), hsa-miR-214-5p (maternal blood; P3) (**c**), and hsa-miR-3177-5p (maternal blood; P5) (**d**). Each RNA sample was validated in triplicate, and the relative expression level of each gene was normalized to that of *RNU6*, which served as the housekeeping gene (*n* = 6). * *p* < 0.05.

**Figure 5 ijms-22-05320-f005:**
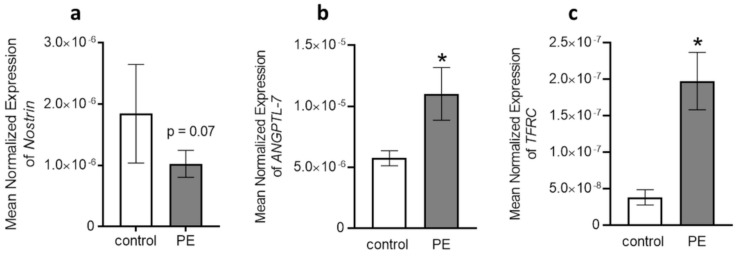
Transcript levels of predicted hsa-miR-214-5p target gene *NOSTRIN* (**a**) and hsa-miR-1270 target genes *ANGPTL7* (**b**) and *TFRC* (**c**) in ECFC at passage 3 from control vs. preeclampsia (PE). Note: * *p* < 0.05, *n* = 6 per group. Each mRNA sample was validated in triplicate and the relative expression level for each was normalized to that of 18S.

**Figure 6 ijms-22-05320-f006:**
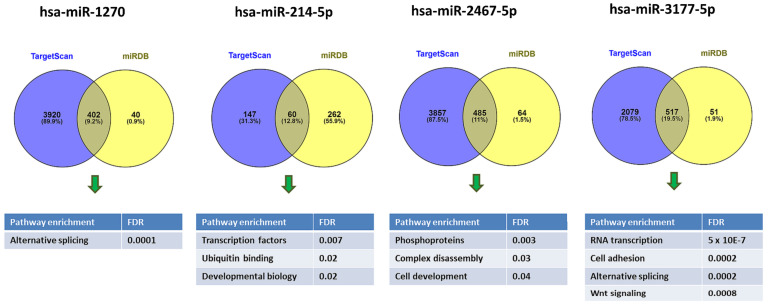
Venn diagram showing the overlap between predicted targets for hsa-miR-1270, hsa-miR-3177-5p, hsa-miR-214-5p, and hsa-miR-2467-5p that were shared by both TargetScan and miRDB prediction tools and were fed into further pathway analyses. Figure generated using Venny 2.1.0 (http://bioinfogp.cnb.csic.es/tools/venny (accessed on 24 July 2019)).

**Figure 7 ijms-22-05320-f007:**
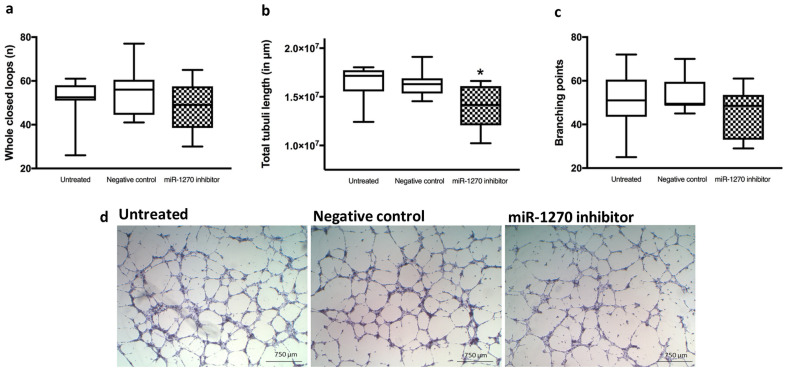
Tube formation of ECFC after inhibition of hsa-miR-1270: number of whole formed loops (**a**); total measured tubule length (**b**); total branching points (**c**); representative microphotographs (**d**); * *p* < 0.05, *n* = 8 per group. Scale bar = 750 µm.

**Figure 8 ijms-22-05320-f008:**
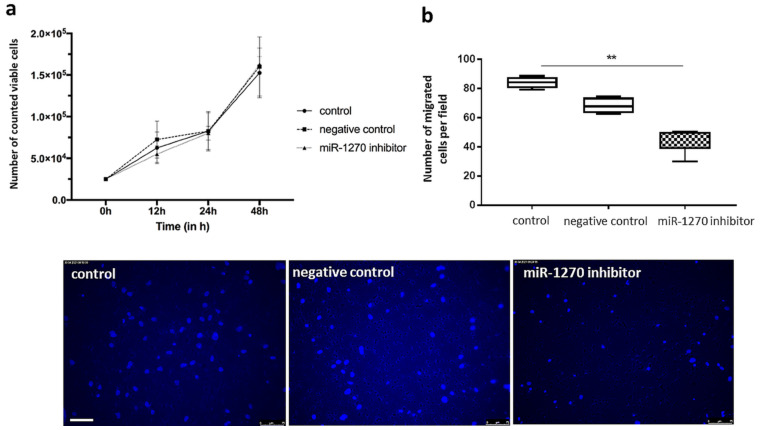
Proliferation (**a**) and chemotactic motility (**b**) of ECFC after inhibition of hsa-miR-1270 compared with negative control. Proliferation: Numbers of counted viable cells at 12, 24, and 48 h; *n* = 4 per group. Chemotactic motility: Numbers of DAPI-stained migrated cells per area were counted in 5 randomly taken pictures from two different biological replicates. Experiments were performed in at least 2–3 technical replicates; ** *p* < 0.01. Scale = 75 µm.

**Table 1 ijms-22-05320-t001:** Patient demographics of cord-blood- and maternal-blood-derived ECFC. Data are expressed as means +/− standard deviation or numbers (*n*) and %. BMI, body mass index; ECFC, endothelial progenitor cells; SBP, systolic blood pressure; DBP, diastolic blood pressure.

	Cord-Blood-Derived ECFC			Maternal-Blood-Derived ECFC		
	Healthy Pregnancy (*n* = 6)	Preeclamptic Pregnancy (*n* = 6)	*p*-Value	Healthy Pregnancy (*n* = 6)	Preeclamptic Pregnancy (*n* = 6)	*p*-Value
Maternal age at delivery (years)	32.8 ± 5.2	31.5 ± 3.7	0.65	31.7 ± 7.4	30.8 ± 5.5	0.85
Gestational age at delivery (weeks)	38.6 ± 0.5	36.8 ± 1.1	0.03	38.3 ± 0.7	33.1 ± 3.9	0.01
Multiparous *n* (%)	5 (83%)	4 (67%)	1.00	6 (100%)	4 (67%)	0.45
Maternal pre-pregnancy BMI (kg/m^2^)	26.2 ± 6.3	29.7 ± 12.9	0.80	27.2 ± 4.5	26.9 ± 6.1	0.90
Gestational SBP, pre-delivery (mmHg)	121 ± 8	164 ± 18	<0.001	121 ± 8	163 ± 11	<0.001
Gestational SBP, before 20 week gestation (mmHg)	112 ± 11	125 ± 21	0.20	116 ± 14	113 ± 16	0.70
Gestational DBP, pre-delivery (mmHg)	70 ± 5	98 ± 6	<0.001	74 ± 9	95 ± 13	0.009
Gestational DBP, before 20 week gestation (mmHg)	69 ± 11	80 ± 10	0.10	68 ± 11	76 ± 9	0.18
Birth weight (g)	3487 ± 523	2741 ± 669	0.06	3314 ± 651	1628 ± 815	0.004
Birth weight percentile	55 ± 30	35 ± 35	0.32	51 ± 35	21 ± 16	0.11
Birth weight percentile <10th *n* (%)	0 (0%)	1 (17%)	1.00	1 (17%)	2 (34%)	1.00
Caesarean delivery *n* (%)	6 (100%)	5 (83%)	1.00	6 (100%)	6 (100%)	1.00
Maternal race, White *n* (%)	6 (100%)	5 (83%)	1.00	6 (100%)	6 (100%)	1.00
Baby sex, Male *n* (%)	3 (50%)	1 (17%)	0.55	1 (17%)	2 (34%)	1.00

**Table 2 ijms-22-05320-t002:** Global overview of regulated miRNAs in preeclamptic ECFC vs. control ECFC with fold changes greater than 2 and *p*-values lower than 0.05. Fold changes refer to the reference group (control = 1).

miRNA	Fold Change	*p*-Value
**ECFC from Cord blood in P3**		
hsa-miR-4726-5p	−2.93	0.03
hsa-miR-1270	−2.63	0.008
hsa-miR-148a-5p	−2.48	0.03
hsa-miR-1255a	−2.06	0.04
**ECFC from Cord blood in P5**		
hsa-miR-2467-5p	3.19	0.005
hsa-miR-4687-3p	2.11	0.03
hsa-miR-148a-3p	−3.44	0.04
hsa-miR-1226-5p	−2.40	0.02
hsa-miR-3911	−2.34	0.007
**ECFC from Maternal blood in P3**		
hsa-miR-214-5p	−10.66	0.002
hsa-miR-199a-5p	−10.45	0.01
hsa-miR-214-3p	−8.89	0.005
hsa-miR-199a-3p	−8.52	0.005
hsa-miR-199b-3p	−8.52	0.005
hsa-miR-139-3p	−8.25	0.009
hsa-miR-139-5p	−4.13	0.04
hsa-miR-551a	−3.91	0.03
hsa-miR-4684-5p	−2.44	0.01
hsa-miR-3115	−2.25	0.04
hsa-miR-3164	−2.16	0.04
hsa-miR-199b-5p	−2.14	0.01
hsa-miR-4728-3 p	5.06	0.003
hsa-miR-1250	3.10	0.04
hsa-miR-338-5p	3.00	0.04
hsa-miR-338-3p	2.92	0.04
hsa-miR-4485	2.75	0.03
hsa-miR-3177-5p	2.27	0.01
hsa-miR-3128	2.21	0.04
hsa-miR-503	2.08	0.04
**ECFC from Maternal blood in P5**		
hsa-miR-214-5p	−7.34	0.03
hsa-miR-214-3p	−6.70	0.02
hsa-miR-199a-3p	−6.26	0.04
hsa-miR-326	−2.70	0.02
hsa-miR-491-3p	−2.09	0.04
hsa-miR-4511	3.45	0.007
hsa-miR-3128	3.18	0.002
hsa-miR-4421	2.41	0.04
hsa-miR-2682-5p	2.23	0.02
hsa-miR-3177-5p	2.19	0.001
hsa-miR-5690	2.08	0.03
hsa-miR-2277-3p	2.06	0.04

**Table 3 ijms-22-05320-t003:** Passage-associated differences in miRNA levels of ECFC from preeclampsia and controls with fold changes greater than 2 and *p*-values lower than 0.05. Fold changes refer to the reference group (P3 = 1).

miRNA	Fold Change	*p*-Value
**Cord blood ECFC from controls in P3 vs. P5**		
hsa-miR-4725-3p	−2.83	0.0007
hsa-miR-1293	−2.44	0.04
hsa-miR-3155a	−2.30	0.03
hsa-miR-548s	−2.19	0.03
hsa-miR-3911	−2.16	0.007
hsa-miR-451a	3.70	0.007
hsa-miR-122-5p	3.20	0.004
hsa-miR-5582-3p	2.36	0.03
hsa-miR-4726-5p	2.14	0.04
hsa-miR-2682-3p	2.10	0.03
**Cord blood ECFC from preeclampsia in P3 vs. P5**		
hsa-miR-1226-5p	−28.23	0.008
hsa-miR-5001-5p	−25.60	0.02
hsa-miR-3911	−21.26	0.02
hsa-miR-3149	20.86	0.008
hsa-miR-4791	20.16	0.04
hsa-miR-2467-5p	3.77	0.002
**Maternal blood ECFC from controls in P3 vs. P5**		
hsa-let-7g-3p	−24.30	0.008
hsa-miR-4421	−2.97	0.003
**Maternal blood ECFC from preeclampsia in P3 vs. P5**		
hsa-miR-4677-5p	−2.45	0.03
hsa-miR-4754	2.22	0.01

## Data Availability

The data that support the findings of this study are available from the corresponding author upon reasonable request.

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
