# Peer review of "MicroRNA Profiles of Maternal and Neonatal Endothelial Progenitor Cells in Preeclampsia"

_ijms, 2021, doi:10.3390/ijms22105320_

Round 1

Reviewer 1 Report

Overall this is a well written manuscript and provides a clear and concise story of how miR-1270 is involved in preeeclampsia. There are some minor concerns that the authors need to address before it should be ready for publication:

  1. General Comment on validation of miRNAs using qPCR: As the authors may know running qPCR for quantifying miRNAs is rather misleading. This is due to the fact that there does not exist any reliable reference or housekeeping miRNA to use for studies. For example in this manuscript the authors utilized RNU6 as their reference miRNA. There have been many publications such the follow: https://www.spandidos-publications.com/10.3892/ijmm.2015.2338#, which show how unreliable this reference actually is. To truly quantify miRNAs a person should be utilizing droplet digital PCR. ddPCR does not require the use reference miRNAs and will produce the exact copies of the miRNAs in your sample. For the manuscript it is recommended that the authors will rerun their qPCR validations utilizing ddPCR for an accurate count of the miRNAs in the blood.
  2. Results: Pg. 2 line 92: State whether this is significant or not.
  3. Results: Pg. 3 Line 100: The authors should state that this is was done by miRNA-seq to be clear on how the results were obtained.
  4. Figure 1: Clearly state either in figure caption or in the figure what each volcano plots is associated with. Currently this is unclear.

Author Response

Dear Mr. Wang,

we thank you and the three reviewers for thoughtful, helpful and detailed comments and suggestions. We appreciate the time taken to help us improve the manuscript. We have addressed each of these comments and accordingly revised our manuscript. Changes to the manuscript are highlighted in the track-changes mode and highlighted as yellow mark. The information on the pages and line numbers in the response letter correspond to those in the deactivated track changes mode.

We hope our revised manuscript is now acceptable for publication.

Sincerely,

Frauke von Versen-Höynck

Reviewer 1:

Overall this is a well written manuscript and provides a clear and concise story of how miR-1270 is involved in preeclampsia. There are some minor concerns that the authors need to address before it should be ready for publication:

  1. General Comment on validation of miRNAs using qPCR: As the authors may know running qPCR for quantifying miRNAs is rather misleading. This is due to the fact that there does not exist any reliable reference or housekeeping miRNA to use for studies. For example in this manuscript the authors utilized RNU6 as their reference miRNA. There have been many publications such the follow: https://www.spandidos-publications.com/10.3892/ijmm.2015.2338#, which show how unreliable this reference actually is. To truly quantify miRNAs a person should be utilizing droplet digital PCR. ddPCR does not require the use reference miRNAs and will produce the exact copies of the miRNAs in your sample. For the manuscript it is recommended that the authors will rerun their qPCR validations utilizing ddPCR for an accurate count of the miRNAs in the blood.

Response from the Authors: We thank the reviewer for these thoughts and suggestions. RNU6 is an established house-keeping gene that has been used in many miRNA studies. Although there is some evidence that RNU6 can show differential expression, as in the article referred to by the reviewer, this can be related to a comparison between malignant and non-malignant cells due to oncogenic transformation. By contrast, our study only uses untransformed cells and compares healthy with preeclamptic pregnancies. The array screen shown by us had not provided evidence that RNU6 is differentially regulated in preeclamptic cells. Moreover, our RT-PCR studies showed very little inter-sample heterogeneity for three of four miRNAs (including the hsa-miR-1270 under question) and consistently confirmed the array data, thereby lending credence to our use of RNU6 as a reliable housekeeper (please see figure 1). In fact, the four miRNAs measured had either lower or higher levels compared to the control group, thereby excluding that the measured effect is due to differential RNU6 levels. A digital droplet PCR would not solve the problem of choosing a house-keeper since a quality control for RNA extraction and cDNA synthesis would still be required.

  1. Results: Pg. 2 line 92: State whether this is significant or not.

Response from the Authors: We thank the reviewer for this helpful comment and have now clearly stated in the manuscript the significance of the results and included p-values in the text (pg. 2, lines 90-94).

  1. Results: Pg. 3 Line 100: The authors should state that this was done by miRNA-seq to be clear on how the results were obtained.

Response from the Authors: We have changed this point in the manuscript and clearly state how the results were obtained (pg. 4, line 128).

  1. Figure 1: Clearly state either in figure caption or in the figure what each volcano plots is associated with. Currently this is unclear.

Response from the Authors: As suggested by the reviewer we characterized Figure 1 in more detail in the figure legend.

Reviewer 2 Report

The article by Lars Brodowski and co-authors investigates in maternal and cord blood-derived endothelial progenitor cells the effect of miRNAs expression alterations in preeclampsia development. The authors analyzed maternal and neonatal miRNAs by RNA-sequencing from endothelial cells of preeclamptic and healthy, control, pregnant individuals. They demonstrate differentially expressed miRNAs, with hsa-miR-1270 showing significantly different levels in cord blood EPCs from preeclampsia versus control samples and this was negatively correlated with mRNA levels of its predicted targets. Functional validation of the effect of hsa-miR-1270 was performed by transfection with a hsa-miR-1270 inhibitor which resulted in decreased tube formation capacity but did not change proliferation in vitro.

1) Labelling of Table1 is rather confusing. The top row includes Cord blood derived ECFC vs maternal blood derived ECFC but the actual table describes primarily patient characteristics and not cells. Also, in the manuscript it is unclear if cord blood derived cells are from the same individual as maternal derived cells. The way table 1 is presented it implies that it is not, therefore making invalid any comparisons between maternal and cord-blood samples.

2) As mentioned in the previous comment the origin of the samples is not clearly mentioned, but I assume that maternal and cord blood samples are taken from the same individual. If so, and since only 6 patients were enrolled to the study, a better analysis on the maternal vs cord blood comparisons would be to compare maternal and cord blood samples of individual patients instead of pooling all maternal and all cord blood samples to compare them between them. This would really help to pinpoint differences between maternal and cord blood miRNAs per case and could also be useful for potential association of expression changes with patient characteristics.

3) The low number of patients included is a limitation of this study and should be mentioned.

4) The majority of RNA-seq expression changes did not pass the Benjamini-Hochberg-correction test of FDR statistical analysis, therefore making their significance questionable. Along these lines, it seems rather strange that although the maternal vs cord blood comparison was the only one to actually give some miRNAs that passed the FDR test (figure 1B), the authors did not choose to follow-up any of these miRNAs. And this is most peculiar keeping in mind the title of the manuscript.

5) It is unclear why the authors chose to study cells at specifically passage 3 and passage 5. Please explain. As shown in Figure 1C, very few miRNAs were alternatively expressed among P3 and P5 in all comparisons, therefore reducing the impact of passage number in the observed effect.

6) Inclusion of Venn diagrams would be useful in depicting the number of common and different miRNA expression changes in the different set of samples. Also, since the authors have performed three different set of comparisons, are there any miRNAs that may be consistently found altered in all of these comparisons?

7) In the methods section the authors describe the different transfection samples, but they should also brief description in the relevant results section so that it is clear to the reader what samples of Figure 5 correspond to. Also, in the methods section, transfection of mirVana™ miRNA inhibitor let-7c positive control is described, however, this is not included in the figure or described in the results section. Please include results obtained from this sample too.

8) Treatment with the inhibitor results in marginal differences, as clearly shown at both the graphs and the representative images provided. Do the authors have additional functional validations on the effect of this inhibitor? Does overexpression of miR-1270 produce the opposite effect? Given the marginal effect of the inhibitor, additional assays would be required to confirm findings.

9) Please include additional description on how measurements of Total tubule length, quantity of whole closed loops and number of interconnections between the tubules (branching points) were performed.

10) Given that alternative splicing is known to be implicated in preeclampsia, it seems unclear as to why the authors did not choose to evaluate this, especially given their findings from the enrichment analysis. Along the same lines, as described in the discussion there are contradictory findings in the bibliography regarding miR-1270 so it seems strange that the authors have chosen this miRNA as their primary finding that was followed up by functional assays.

11) Previous studies have already identified miRNA expression changes in plasma from preeclampsia vs control samples. Do the authors detect any of these miRNA changed in their experimental set up?

12) In discussion (line 298, page 12) the authors propose that ‘miRNAs in each group may synergize in the regulation of gene expression’, however there is no evidence towards this. Keeping in mind that the majority of changes are of low statistical significance, then this is actually highly improbable.  

13) Figures 1A, 1B and 1C should be enlarged so that they are easier to read.

Author Response

Dear Mr. Wang,

we thank you and the three reviewers for thoughtful, helpful and detailed comments and suggestions. We appreciate the time taken to help us improve the manuscript. We have addressed each of these comments and accordingly revised our manuscript. Changes to the manuscript are highlighted in the track-changes mode and highlighted as yellow mark. The information on the pages and line numbers in the response letter correspond to those in the deactivated track changes mode.

We hope our revised manuscript is now acceptable for publication.

Sincerely,

Frauke von Versen-Höynck

Reviewer 2:

The article by Lars Brodowski and co-authors investigates in maternal and cord blood-derived endothelial progenitor cells the effect of miRNAs expression alterations in preeclampsia development. The authors analyzed maternal and neonatal miRNAs by RNA-sequencing from endothelial cells of preeclamptic and healthy, control, pregnant individuals. They demonstrate differentially expressed miRNAs, with hsa-miR-1270 showing significantly different levels in cord blood EPCs from preeclampsia versus control samples and this was negatively correlated with mRNA levels of its predicted targets. Functional validation of the effect of hsa-miR-1270 was performed by transfection with a hsa-miR-1270 inhibitor which resulted in decreased tube formation capacity but did not change proliferation in vitro.

1) Labelling of Table1 is rather confusing. The top row includes Cord blood derived ECFC vs maternal blood derived ECFC but the actual table describes primarily patient characteristics and not cells. Also, in the manuscript it is unclear if cord blood derived cells are from the same individual as maternal derived cells. The way table 1 is presented it implies that it is not, therefore making invalid any comparisons between maternal and cord-blood samples.

Response from the Authors: We agree with the reviewer on this point and thank for the valuable suggestions. We have now re-ordered Table 1 and further have included a subsection in the manuscript (Section 2.2., pg. 3 lines 103-114) and a separate figure about the data of ECFC colony numbers and appearance (Suppl. Figure S1).Further, we provide more information in the methods section (pg. 15, lines 452-456) about the composition of the study groups and revised the discussion (pg. 14, lines 412-420). Within the healthy pregnancies (n = 9) we were able to isolate maternal as well as cord blood ECFC for three mother-child pairs. Unfortunately, there were no pairs available to us from preeclamptic patients (n = 12). This is due to the difficult cell isolation especially from adult blood which contains only a hundredth of ECFC compared to cord blood. We believe the comparison of unrelated maternal and cord blood ECFC is still a good starting point while we acknowledge that it would be desirable to compare pairs of maternal and cord blood ECFC of the same individual in the future.

2) As mentioned in the previous comment the origin of the samples is not clearly mentioned, but I assume that maternal and cord blood samples are taken from the same individual. If so, and since only 6 patients were enrolled to the study, a better analysis on the maternal vs cord blood comparisons would be to compare maternal and cord blood samples of individual patients instead of pooling all maternal and all cord blood samples to compare them between them. This would really help to pinpoint differences between maternal and cord blood miRNAs per case and could also be useful for potential association of expression changes with patient characteristics.

Response from the Authors: Please see our response to your previous question.

3) The low number of patients included is a limitation of this study and should be mentioned.

 Response from the Authors: We have now clearly stated the limitation of the study in the discussion (pg. 14, lines 412-420). Yet, we would like to point out that miRNA studies on circulating miRNA [1] from cord blood as well as from cord-blood derived ECFC [2] have been performed with relatively low numbers of samples per group (n=3-4). At least for the primary cells, e.g. ECFC the limitation arises from a limited isolation efficacy which is associated with lower numbers of available cell samples in our and other studies [2].

4) The majority of RNA-seq expression changes did not pass the Benjamini-Hochberg-correction test of FDR statistical analysis, therefore making their significance questionable. Along these lines, it seems rather strange that although the maternal vs cord blood comparison was the only one to actually give some miRNAs that passed the FDR test (figure 1B), the authors did not choose to follow-up any of these miRNAs. And this is most peculiar keeping in mind the title of the manuscript.

 Response from the Authors: We agree with the reviewer that this is an important point. Although some miRNAs passed the Benjamini-Hochberg correction test in cord blood vs maternal blood (preeclamptic) comparison, we decided not to focus further analysis on this group. This is because we would otherwise have conducted an analysis within the same entity (preeclampsia). The main objective of our manuscript was to explore differences in miRNA expression in the preeclamptic vs. control groups, divided into cord blood and maternal blood ECFC. Furthermore, another focus of the analysis was to investigate the different miRNA levels within different cell culture passages. Therefore, we chose to further validate a set of 4 miRNAs that showed differences between preeclampsia cases and controls using the lowest p-value in decision-making. Also, since we had few maternal-cord blood pairs available we didn’t follow up on this comparison at this point. We revised the title of the manuscript to avoid misunderstanding.

5) It is unclear why the authors chose to study cells at specifically passage 3 and passage 5. Please explain. As shown in Figure 1C, very few miRNAs were alternatively expressed among P3 and P5 in all comparisons, therefore reducing the impact of passage number in the observed effect.

Response from the Authors: We agree with the reviewer that this point was somewhat not clearly stated. We now included more precisely an explanation in the methods (pg. 16, lines 512-515) and in the discussion (pg. 12, lines 347-352).

One major aim of our study was to determine if ECFC show „developmental“ regulation of miRNA profiles over cell culture passages. We have used P3 for this analysis, as this passage reflects the passage that provides the required quantity and quality for the analyses used. Since ECFC are primary cells and are therefore limited in their cell proliferation and from passages 7-8 onwards show further morphological and molecular changes due to their ageing process, we deliberately chose P5 as second passage for a comparison. P5 offered us the possibility to analyse the developmental change of the miRNA profile in more detail and to avoid possible ageing processes due to limited proliferation in later passages. Further, the result shows that the priming of endothelial progenitor cells obtained in each individual (by the different factors of preeclampsia or in a healthy patient) has a largely consistent influence on the miRNA profile in the continuing development of the cells. This is also an important information for us for further investigations of the primary ECFC used, as we can now exclude different cellular effects due to strongly altered profiles within the different number of passages.

6) Inclusion of Venn diagrams would be useful in depicting the number of common and different miRNA expression changes in the different set of samples. Also, since the authors have performed three different set of comparisons, are there any miRNAs that may be consistently found altered in all of these comparisons?

 Response from the Authors: We thank the reviewer for this suggestion and created 4 Venn diagrams for the respective groups (Supplemental Figures S4-7 and results: pg. 4, lines 121-125, discussion: pg. 13, lines 358-360). Two miRNAs showed different expression changes in all entities (hsa-miR-2467-5p; hsa-miR-4421).

7) In the methods section the authors describe the different transfection samples, but they should also brief description in the relevant results section so that it is clear to the reader what samples of Figure 5 correspond to. Also, in the methods section, transfection of mirVana™ miRNA inhibitor let-7c positive control is described, however, this is not included in the figure or described in the results section. Please include results obtained from this sample too.

 Response from the Authors: We thank the reviewer for this important note. We have now included an explanation of the materials and samples used in the results section (pg. 10, lines 281-287) and show the data for the positive control in a supplemental figure (Supplemental Figure S3).

8) Treatment with the inhibitor results in marginal differences, as clearly shown at both the graphs and the representative images provided. Do the authors have additional functional validations on the effect of this inhibitor? Does overexpression of miR-1270 produce the opposite effect? Given the marginal effect of the inhibitor, additional assays would be required to confirm findings.

Response from the Authors: During the limited time allowed to resubmit the revision, we carried out further functional analysis of the hsa-miR-1270 inhibitor as suggested by the reviewer. We determined the chemotactic motility of ECFC after miR-1270 inhibition (methods: pg.17, lines 578-591, results: pg. 11, lines 314-318, Figure 8 (II)). Due to delivery problems of the company of the hsa-miR-1270 inhibitor, we were able to explore 2 biological cell lines (ECFC) with 2-3 technical replicates each.

9) Please include additional description on how measurements of Total tubule length, quantity of whole closed loops and number of interconnections between the tubules (branching points) were performed.

 Response from the Authors: We appreciate this suggestion. We included an additional description in the methods section on how measurements were performed (pg. 17, lines 565-571).

10) Given that alternative splicing is known to be implicated in preeclampsia, it seems unclear as to why the authors did not choose to evaluate this, especially given their findings from the enrichment analysis. Along the same lines, as described in the discussion there are contradictory findings in the bibliography regarding miR-1270 so it seems strange that the authors have chosen this miRNA as their primary finding that was followed up by functional assays.

Response from the Authors: We appreciate this important comment. We are pleased to announce that we just recently started a follow-up project where we will investigate the impact of the alternative splice in depth. This project will become another focus of our work for the next years and will help validate the data obtained so far.

We agree with the reviewer that it was not clearly described in our manuscript why we chose miR-1270 for further analysis. First, after determining the 4 miRNAs with the lowest p-value, we decided to use only miRNAs that showed variance in the profile in P3 for further analysis. Since we did not find much difference in miRNA profiles between P3 and P5, we chose to study only P3 cells, as they most closely resemble the phenotype of freshly isolated cells and are therefore more comparable to cells in vivo than primary cells in P5.

Further, we performed an analysis of predicted target genes of miR-214-5p (NOSTRIN) and miR-1270 (TFRC). Since we could not find any differences in the analysis of NOSTRIN for miR-214-5p between preeclampsia and control we included another target gene for miR-1270 (ANGPTL7), where we detected differences in qRT-pCR for both target genes. We selected TFRC because it had the highest score for miR-1270 on TargetScan and ANGPTL-7 because it is related to angiogenesis and thus has a potentially important link to preeclampsia. Therefore, we analysed this miRNA further in functional assays and not miR-214-5p. We have now included the results of miR-214-5p in the results section and describe more clearly why we selected miR-1270 (see pg. 9, lines 232-243).

11) Previous studies have already identified miRNA expression changes in plasma from preeclampsia vs control samples. Do the authors detect any of these miRNA changed in their experimental set up?

 Response from the Authors: Studies carried out to date searching for significantly altered miRNAs between preeclamptic and healthy pregnant women were mainly performed in placenta or plasma but never before in endothelial progenitor cells. Two miRNAs were differently expressed between preeclampsia and controls in other studies as well as in ours. That applies to a downregulation of miR-139-5p in placenta [3] as well as in maternal ECFC at P3 and for a downregulation of miR-148a in decidua-derived mesenchymal stem cells [4] and cord blood-derived ECFC in our study. The reason for the small overlap between previous observations and ours is most likely the different material studied. We have included this information in the discussion (pg. 12, lines 339-342).

12) In discussion (line 298, page 12) the authors propose that ‘miRNAs in each group may synergize in the regulation of gene expression’, however there is no evidence towards this. Keeping in mind that the majority of changes are of low statistical significance, then this is actually highly improbable.

Response from the Authors: Thanks for pointing out this valid point. We have now deleted the sentence in the discussion.

13) Figures 1A, 1B and 1C should be enlarged so that they are easier to read.

 Response from the Authors: We have now enlarged the figures and added a new numbering (Fig.1-3)

Reviewer 3 Report

The authors report an interesting study on miRNA profiles in preeclampsia, in which the authors compared healthy controls with subjects with preeclampsia. Furthermore, cord blood- and maternal blood-derived endothelial colony forming cells (ECFCs) were studied at different cell passages. Overall, it is an important contribution to the field.

However, major revisions are needed before consideration for publication:

1. Study design and Participants:

a) The most unclear point is how many female subjects have been enrolled in this study. In the methods section, no clear information is provided on this. First, the authors state "We recruited subjects (...)" (line 349, page 13) and further below state that "the global miRNA profiles of maternal blood and cord-blood derived" ECFCs of six preeclamptic pregnancies and six uncomplicated healthy pregnancies were determined by small RNA-sequencing" (lines 369-371, page 13). However, when analysing Table 1 (page 3), a difference is observed in the baseline characteristics of the healthy pregnancy and preeclamptic pregnancy groups between cord blood- and maternal blood-derived ECFCs. The authors need to clarify this point, as differences were observed between cord blood- and maternal blood-derived ECFCs. It would have been useful to compare cord blood- and maternal blood-derived ECFCs from the same subjects.

b) Also, the text should reflect such clarification. To this point, the authors stated in the results section that "Gestational age and birth weight were lower in pregnancies complicated by preeclampsia while diastolic and systolic blood pessure was higher" (lines 91-92, page 2). Although a lower birth weight was indeed observed in both types of ECFCs, the statement does not hold entirely true, as the difference in birth weight was only statistically significant in maternal blood derived ECFCs.

c) A major question here is also the definition of preeclampsia. In the methods, the authors state "We used the current definition for preeclampsia of the International Society for the Study of Hypertension in Pregnancy (...)" (lines 353-355, page 13). It would be beneficial to include the definition here.

2. Demographic characteristics:

a) The authors included time to first ECFC colonies and total number of ECFC colonies as patient demographics. This does not seem to be adequate. In fact, these data are results from this study regarding ECFC isolation (as detailed in the methods section). Therefore, I would suggest the authors revise the results by creating a subsection on "ECFC isolation and characterization" where they include these data as well as data from Supplemental Figure S1 in a brief way.

b) Another question on demographic characteristics is how did the authors measure diastolic and systolic blood pressure. It appears no information was provided on this in the methods section.

3. Introduction: The last paragraph could be split into 2 paragraphs: one contextualizing miRNAs and one dedicated to the aim of the manuscript. At the moment, these are merged together, which does not facilitate the reading.

4. Figures and Tables:

a) The authors should re-think Figures 1A, 1B and 1C: either (i) as one figure with 1A, 1B and 1C panels or (ii) as 3 separate figures (1, 2, 3). In the case the authors opt for the first (i.e. Figure 1 divided into 1A, 1B and 1C panels) the authors should consider a different way of presenting the results in table form in each figure (scheme as stated by the authors in the text) and should clearly identify the comparisons being made, i.e. controls vs preeclampsia (1A), cord blood vs maternal blood (1B) and P3 vs P5 (1C).

b) Table 1: Replace +/- with the symbol ± throughout the table.

c) Table 2: This table should be re-drawn in order to improve readability of the data: (i) there is no need to indicate several times that the data refer to the comparison between control and preeclampsia (this indication in the caption of the table will be enough); (ii) the authors ordered miRNAs according to the P-value, however grouping by regulation, i.e. down and upregulation, and only then by P-value could provide a better picture; (iii) presenting fold change as positive and negative values (as in Table 3) would indicate the same information as a column on up- or downregulation and a column with the value of fold change, but in a simpler way, so the authors should consider this option (in this case, the "regulation" column would be unnecessary).

c) Table 3:

i. In the text, the authors state in section 2.3 that "the most strongly regulated miRNAs (...) can be found in Supplemental Table S1" (lines 131-132, page 5) and in section 2.4 that "A list of miRNAs with more than two-fold changes (...) between P3 and P5 is provided in Table 3" (lines 152-153, page 7). It seems these have been switched, i.e. Table 3 seems to refer to cord blood vs maternal blood-derived ECFCs independently of passage as Supplemental Table S1 refers to cord blood vs maternal blood-derived ECFCs taking into consideration passage (P3 and P5). The authors need to correct this.

ii. Having +/- signals next to fold change values, makes the "regulation" column unnecessary. Similarly to above, there is no need to indicate several times that the data refer to the comparison between cord blood and maternal blood (this indication in the caption of the table for example will be enough) and also grouping by regulation, i.e. down and upregulation, and only then by P-value could also provide a better picture. The same applies to Supplemental Table S1.

d) In the text, authors refer to schemes I-IV from figures 1A-1C. Do they mean panels or graphs? Clarifying the issue on the numbering and adjustment of these figures as previously stated, will allow to better represent these graphs and their organization. The authors must try to improve this.

5. miRNA profiles in the preeclampsia and control groups (section 2.2):

a) The authors could include a statement, in the previous section, that the miRNA profile was compared first between preeclampsia and control groups, then between maternal blood and cord blood and ultimately between passages 3 and 5 (P3 and P5, respectively). This would add a bit more context and afterwards each subsection would be dedicated to each comparison.

b) The set of statistical criteria that were used to carry out this comparison included a moderated t-test (P lower or equal than 0.05) and a fold change > 2. First, no particular reason is given to consider P lower or equal than 0.05, when the majority of the literature uses p < 0.05 (at least the authors could give an explanation). Second, the authors do not mention in the methods section that a threshold of 2 was used for the fold change, as they only state "the biggest fold change (...)" (line 414, page 14). Third, the caption of Table 2 state P-value less than 0.05. The authors should revise the manuscript in regard to this issue.

c) Regarding Supplemental Figure 2A, the authors show heat maps with expression changes of candidate miRNAs. However, it is unclear what this means. Do the heat maps refer to data from each subject in the study? If so, why are there 7 subjects in maternal blood-derived ECFC group with preeclampsia in 1A II and only 5 in 1A IV, when there are 6 subjects in each group? Only 5 values are presented in 1C IV of supplementary figure 2C, as well. Also, how are these values of fold change calculated, before 20-week gestation vs pre-delivery or other criterion? In addition, what is the meaning of fold change in the Tables, does it refer to the mean fold change between groups? Overall, the authors need to clearly indicate what the fold change refers to in the several figures and tables, as this is fundamental to interpret the results of the study.

6. miRNA differences of cell culture passages 3 and 5 (section 2.4): In lines 153-154 (page 7), the authors state "only one miRNA was shared between preeclampsia and healthy controls". The authors should indicate which miRNA are they referring to, as well as the changes that were observed.

7. Quantitative RT-PCR validation and putative target gene levels (section 2.5):

a) Did the authors consider any threshold when assessing the miRNAs with the most significant difference, i.e. lowest P-value? (line 158, page 7). Corresponding information should also be included in the respective methods subsection.

b) Fold change values presented in lines 160-162 (page 7) should include a signal to better represent if up- or downregulation was observed.

c) The authors "selectively explored whether the levels of ANGPTL7 or TFRC correlated with hsa-miR-1270 in cord blood ECFCs" (lines 171-172, page 8). The authors should include an explanation to why they only explored these.

8. Tube formation and proliferation (section 2.7): Why did the authors only explored hsa-miR-1270 in this analysis, when other miRNAs were also down- (hsa-miR-2467-5p) or up-regulated (hsa-miR-214-5p and hsa-miR-3177-5p)? The authors should include an explanation to this.

9. Discussion:

a) First, I would suggest the authors do not fragment the discussion into topics/subsections, as it is unnecessary. A coherent flow of information does not require subsections in the discussion.

b) The authors state "we focused on hsa-miR-1270 as it was the most different miRNA (...)" (lines 238-239, page 10). What do the authors mean by "most different"? The authors considered only the P-value (as detailed in lines 158-159, page 7), but did not base that decision on a measure of change (i.e. fold change), therefore this statement is somehow arguable. In fact, when observing Table 2, a higher fold change is seen for hsa-miR-4726-5p (-2.93 vs -2.63 in hsa-miR-1270), even though the P-value was higher (P=0.03 compared to P=0.008 in hsa-miR-1270). Overall, the authors should improve the justification for this decision throughout the manuscript.

c) The authors point out that "previous studies on plasma and placental miRNAs have indicated the involvement of miRNAs in the pathophysiology of preeclampsia" (lines 245-247, pages 10-11) and also that "there have been few overlap among these studuies and more investigation is required in order to clarify the exact role of preeclampsia-related miRNAs" (lines 249-250, page 11). As multiple miRNAs have been implicated in preeclampsia, it would be useful to further discuss the implications of the body of evidence that has been accumulating on this topic - suggest consultation of Lv et al. 2019 (J Cell Physiol, 234:1052–1061) and Apicella et al. 2019 (Int J Mol Sci, 20:2837). A discussion on the specific role of these miRNAs would add perspective to these observations.

d) In lines 272-273 (page 11), the authors state that only two studies have investigated the functional role of hsa-miR-1270. However, the authors do not include the bibliographic references of such studies. This is essential. Also, I would suggest to check and revise reference citation throughout the text.

10. Methods:

a) In line 383 (page 13), "a detailed description is provided as supplemental material". In the final sentence of the same paragraph, the authors state this in a similar way, thus I would suggest to revise into one single sentence.

b) In line 391 (page 13), the authors mention an alpha level of 0.1 for normality tests. Why did the authors choose this threshold instead of the common threshold of P<0.05, which indeed was used later on moderated t-test? The authors should include an explanation to this.

c) The sentence in lines 399-400 (page 14) should be revised in regard to phrasing.

d) The authors used an hsa-miR-1270 inhibitor (lines 449, page 15), but do not state which compound/inhibitor. Similarly, the negative control in cell proliferation assays (line 462, page 15) should be defined, was it the vehicle?

e) Statistical analysis: which test was used to assess the normality of distribution, Kolmogorov-Smirnov (lines 465-466, page 15), Shapiro-Wilk (lines 389-390, page 13) or both? This should be clarified.

f) The last sentence of the statistical analysis subsection needs revision, as it begins with a URL.

Additional points:

L. 73, "preeclampsia (51)" (page 2) - Does (51) correspond to a reference? If so, which reference? (reference list ends with 48 references).

L. 110, "some 47 miRNAs" (page 4) - Remove "some".

L. 113, "some 39 miRNAs" (page 4) - Remove "some".

L. 272, "hitherto" (page 11) - The authors could find a more common word for such meaning, as it is an unusual term in scientific literature.

L. 402, "passage 3 (P3) and passage 5" (page 14) - Revise into abbreviations, i.e. P3 and P5 (these have been previously defined in the text).

L. 414, "fold change (FC)" (page 14) - This abbreviation is not necessary (only appears in schemes inside Figures 1A, 1B and 1C); also it could create confusion with ECFC.

L. 479-482 (page 15) - Check text formatting.

In Supplemental Table S3A, the header states "Sequenz", please revise it.

Author Response

Dear Mr. Wang,

we thank you and the three reviewers for thoughtful, helpful and detailed comments and suggestions. We appreciate the time taken to help us improve the manuscript. We have addressed each of these comments and accordingly revised our manuscript. Changes to the manuscript are highlighted in the track-changes mode and highlighted as yellow mark. The information on the pages and line numbers in the response letter correspond to those in the deactivated track changes mode.

We hope our revised manuscript is now acceptable for publication.

Sincerely,

Frauke von Versen-Höynck

Reviewer 3:

The authors report an interesting study on miRNA profiles in preeclampsia, in which the authors compared healthy controls with subjects with preeclampsia. Furthermore, cord blood- and maternal blood-derived endothelial colony forming cells (ECFC) were studied at different cell passages. Overall, it is an important contribution to the field.

However, major revisions are needed before consideration for publication:

  1. Study design and Participants:
  2. a) The most unclear point is how many female subjects have been enrolled in this study. In the methods section, no clear information is provided on this. First, the authors state "We recruited subjects (...)" (line 349, page 13) and further below state that "the global miRNA profiles of maternal blood and cord-blood derived" ECFC of six preeclamptic pregnancies and six uncomplicated healthy pregnancies were determined by small RNA-sequencing" (lines 369-371, page 13). However, when analysing Table 1 (page 3), a difference is observed in the baseline characteristics of the healthy pregnancy and preeclamptic pregnancy groups between cord blood- and maternal blood-derived ECFC. The authors need to clarify this point, as differences were observed between cord blood- and maternal blood-derived ECFC. It would have been useful to compare cord blood- and maternal blood-derived ECFC from the same subjects.

 Response from the Authors: We agree that this was not stated clearly in the manuscript. We now clarified the number of participants in each group in the methods section (pg. 15, lines 452-456). In total, we isolated ECFC from 12 women with preeclampsia and from 9 with healthy pregnancies. Within the healthy pregnancies we were able to isolate maternal as well as cord blood ECFC for three mother-child pairs. Unfortunately, there were no pairs available to us from preeclamptic patients. The isolation of ECFC is challenging and especially from adult blood the efficiency is below 50% which can be explained by lower cell numbers (e.g. only a hundredth of ECFC in adult blood compared to cord blood). We hope to be able to perform a broader analysis with more pairs for comparison in future projects.

  1. b) Also, the text should reflect such clarification. To this point, the authors stated in the results section that "Gestational age and birth weight were lower in pregnancies complicated by preeclampsia while diastolic and systolic blood pressure was higher" (lines 91-92, page 2). Although a lower birth weight was indeed observed in both types of ECFC, the statement does not hold entirely true, as the difference in birth weight was only statistically significant in maternal blood derived ECFC.

 Response from the Authors: We thanks the reviewer for catching this mistake and revised the paragraph accordingly (pg. 2, lines 90-94).

  1. c) A major question here is also the definition of preeclampsia. In the methods, the authors state "We used the current definition for preeclampsia of the International Society for the Study of Hypertension in Pregnancy (...)" (lines 353-355, page 13). It would be beneficial to include the definition here.

 Response from the Authors: We followed the recommendation of the reviewer and included the definition of preeclampsia in the text as follows: “Preeclampsia was defined as new onset hypertension ³140/90 mmHg on two or more occasions after 20 weeks gestation accompanied by proteinuria and/or other symptoms of organ dysfunction; e.g. liver or acute kidney dysfunction, neurological symptoms, hemolysis, thrombocytopenia or fetal growth restriction” (pg. 15, lines 457-461).

  1. Demographic characteristics:
  2. a) The authors included time to first ECFC colonies and total number of ECFC colonies as patient demographics. This does not seem to be adequate. In fact, these data are results from this study regarding ECFC isolation (as detailed in the methods section). Therefore, I would suggest the authors revise the results by creating a subsection on "ECFC isolation and characterization" where they include these data as well as data from Supplemental Figure S1 in a brief way.

Response from the Authors: We followed the recommendation of the reviewer and included a section in the results part and an additional Figure (Supplemental Figure S1) where we included these data (section 2.2, pg. 3, lines 103-114).

  1. b) Another question on demographic characteristics is how did the authors measure diastolic and systolic blood pressure. It appears no information was provided on this in the methods section.

Response from the Authors: We added the following information on blood pressure measurement to the methods section (section: participants): “Resting blood pressures were taken by a trained nurse using the oscillometric method and the appropriate cuff size.“ (pg. 15, line 461-462).

  1. Introduction: The last paragraph could be split into 2 paragraphs: one contextualizing miRNAs and one dedicated to the aim of the manuscript. At the moment, these are merged together, which does not facilitate the reading.

 Response from the Authors: As suggested we split the last paragraph into 2 (pg. 2, lines 64-84).

  1. Figures and Tables:
  2. a) The authors should re-think Figures 1A, 1B and 1C: either (i) as one figure with 1A, 1B and 1C panels or (ii) as 3 separate figures (1, 2, 3). In the case the authors opt for the first (i.e. Figure 1 divided into 1A, 1B and 1C panels) the authors should consider a different way of presenting the results in table form in each figure (scheme as stated by the authors in the text) and should clearly identify the comparisons being made, i.e. controls vs preeclampsia (1A), cord blood vs maternal blood (1B) and P3 vs P5 (1C).

 Response from the Authors: We have followed the recommendation and have now enlarged the pictures and added a new numbering (Fig.1-3).

  1. b) Table 1: Replace +/- with the symbol ± throughout the table.

 Response from the Authors: We replaced the symbol as suggested.

  1. c) Table 2: This table should be re-drawn in order to improve readability of the data: (i) there is no need to indicate several times that the data refer to the comparison between control and preeclampsia (this indication in the caption of the table will be enough); (ii) the authors ordered miRNAs according to the P-value, however grouping by regulation, i.e. down and upregulation, and only then by P-value could provide a better picture; (iii) presenting fold change as positive and negative values (as in Table 3) would indicate the same information as a column on up- or downregulation and a column with the value of fold change, but in a simpler way, so the authors should consider this option (in this case, the "regulation" column would be unnecessary).

 Response from the Authors: We thank the reviewer for this valuable suggestion. We have now re-ordered the table and ordered by regulation of the fold change. Further, we made an indication of the comparison made in the caption of the table.

  1. c) Table 3:
  2. In the text, the authors state in section 2.3 that "the most strongly regulated miRNAs (...) can be found in Supplemental Table S1" (lines 131-132, page 5) and in section 2.4 that "A list of miRNAs with more than two-fold changes (...) between P3 and P5 is provided in Table 3" (lines 152-153, page 7). It seems these have been switched, i.e. Table 3 seems to refer to cord blood vs maternal blood-derived ECFC independently of passage as Supplemental Table S1 refers to cord blood vs maternal blood-derived ECFC taking into consideration passage (P3 and P5). The authors need to correct this.

 Response from the Authors: We thank the reviewer for this comment. We proofed and revised the table labeling and order.

  1. Having +/- signals next to fold change values, makes the "regulation" column unnecessary. Similarly to above, there is no need to indicate several times that the data refer to the comparison between cord blood and maternal blood (this indication in the caption of the table for example will be enough) and also grouping by regulation, i.e. down and upregulation, and only then by P-value could also provide a better picture. The same applies to Supplemental Table S1.

 Response from the Authors: We have re-ordered Table 3 and Supplemental Table 1 by regulation of the fold change and we provide an indication of the comparison in the caption of the table.

  1. d) In the text, authors refer to schemes I-IV from figures 1A-1C. Do they mean panels or graphs? Clarifying the issue on the numbering and adjustment of these figures as previously stated, will allow to better represent these graphs and their organization. The authors must try to improve this.

Response from the Authors: We agree with the reviewer and have re-numbered the figures, adjusted the size and gave a more specific description in the figure legend.

  1. miRNA profiles in the preeclampsia and control groups (section 2.2):
  2. a) The authors could include a statement, in the previous section, that the miRNA profile was compared first between preeclampsia and control groups, then between maternal blood and cord blood and ultimately between passages 3 and 5 (P3 and P5, respectively). This would add a bit more context and afterwards each subsection would be dedicated to each comparison.

 Response from the Authors: We thank the reviewer for this suggestion and have now included a section on our three main objectives in the manuscript before we show the group related results (pg. 4, lines 117-125).

  1. b) The set of statistical criteria that were used to carry out this comparison included a moderated t-test (P lower or equal than 0.05) and a fold change > 2. First, no particular reason is given to consider P lower or equal than 0.05, when the majority of the literature uses p < 0.05 (at least the authors could give an explanation). Second, the authors do not mention in the methods section that a threshold of 2 was used for the fold change, as they only state "the biggest fold change (...)" (line 414, page 14). Third, the caption of Table 2 state P-value less than 0.05. The authors should revise the manuscript in regard to this issue.

 Response from the Authors: We removed the 'equal than' in the manuscript. The data, filtering or candidate lists were not affected by this change. The caption of Table 2 was left unchanged accordingly. We integrated the filter step “2-fold” into the method part (pg. 16, line 517-520). Indeed, this filter step was not carried out for the display of the Volcano plots to show all affected values, but applies to the candidates in the tables to provide a better overview.

  1. c) Regarding Supplemental Figure 2A, the authors show heat maps with expression changes of candidate miRNAs. However, it is unclear what this means. Do the heat maps refer to data from each subject in the study? If so, why are there 7 subjects in maternal blood-derived ECFC group with preeclampsia in 1A II and only 5 in 1A IV, when there are 6 subjects in each group? Only 5 values are presented in 1C IV of supplementary figure 2C, as well. Also, how are these values of fold change calculated, before 20-week gestation vs pre-delivery or other criterion? In addition, what is the meaning of fold change in the Tables, does it refer to the mean fold change between groups? Overall, the authors need to clearly indicate what the fold change refers to in the several figures and tables, as this is fundamental to interpret the results of the study.

 Response from the Authors: We thank the reviewer for this advice. We revised the heat maps as there are rightly 6 subjects per group as presented now. Each heatmap shows the significantly changed miRNAs with a fold change greater 2. The fold change term applies to comparisons between the reference group and the “cases”. As commented above figures and tables were revised as suggested by the reviewer.

  1. miRNA differences of cell culture passages 3 and 5 (section 2.4): In lines 153-154 (page 7), the authors state "only one miRNA was shared between preeclampsia and healthy controls". The authors should indicate which miRNA are they referring to, as well as the changes that were observed.

 Response from the Authors: We thank the reviewer for this comment. We now named the specific miRNA (hsa-miR-3911) in the text and provide the changes observed as well (pg. 7, lines 193-197).

  1. Quantitative RT-PCR validation and putative target gene levels (section 2.5):
  2. a) Did the authors consider any threshold when assessing the miRNAs with the most significant difference, i.e. lowest P-value? (line 158, page 7). Corresponding information should also be included in the respective methods subsection.

 Response from the Authors: Information on the filtering threshold is provided in the methods section (pg. 16, lines 517-519).

  1. b) Fold change values presented in lines 160-162 (page 7) should include a signal to better represent if up- or downregulation was observed.

 Response from the Authors: Fold change values are now presented including a signal to represent up and downregulation (pg. 8, lines 221-223).

  1. c) The authors "selectively explored whether the levels of ANGPTL7 or TFRC correlated with hsa-miR-1270 in cord blood ECFC" (lines 171-172, page 8). The authors should include an explanation to why they only explored these.

 Response from the Authors: We agree with both reviewers that it was not clearly described why we chose miR-1270 for further analysis. First, after determining the 4 miRNAs with the lowest p-value, we decided to use only miRNAs that showed variance in the profile in P3 for further analysis. Although we did not find much difference in miRNA profiles between P3 and P5, we chose to study only P3 cells, as they most closely resemble the phenotype of freshly isolated cells and are therefore more comparable to cells in vivo than cells in P5. Furthermore, we also performed an analysis of predicted target genes of miR-214 -5p and miR-1270. Since we could not find any differences in the analysis of Nostrin for miR-214-5p between preeclampsia vs. control we analysed target genes for miR-1270 (ANGPTL7 and TFRC). TFRC was chosen because it had the highest score for miR-1270 on TargetScan. ANGPTL7 was chosen because it is related to angiogenesis and thus has an important link to pre-eclampsia. Since we found significant higher ANGPTL7 and TFRC mRNA levels in cord blood ECFC from preeclampsia compared to control, we analysed miR-1270  further using functional assays. We have now also included the results of miR-214-5p in the results section and describe more clearly why we selected miR-1270 (pg. 9, lines 232-243).

  1. Tube formation and proliferation (section 2.7): Why did the authors only explored hsa-miR-1270 in this analysis, when other miRNAs were also down- (hsa-miR-2467-5p) or up-regulated (hsa-miR-214-5p and hsa-miR-3177-5p)? The authors should include an explanation to this.

 Response from the Authors: We have stated an explanation in the previous comment (7c).

  1. Discussion:
  2. a) First, I would suggest the authors do not fragment the discussion into topics/subsections, as it is unnecessary. A coherent flow of information does not require subsections in the discussion.

 Response from the Authors: We have deleted the subsection titles from the discussion.

  1. b) The authors state "we focused on hsa-miR-1270 as it was the most different miRNA (...)" (lines 238-239, page 10). What do the authors mean by "most different"? The authors considered only the P-value (as detailed in lines 158-159, page 7), but did not base that decision on a measure of change (i.e. fold change), therefore this statement is somehow arguable. In fact, when observing Table 2, a higher fold change is seen for hsa-miR-4726-5p (-2.93 vs -2.63 in hsa-miR-1270), even though the P-value was higher (P=0.03 compared to P=0.008 in hsa-miR-1270). Overall, the authors should improve the justification for this decision throughout the manuscript.

 Response from the Authors: We followed the most significant hit, independent of its effect size, which is a common procedure in case-control association studies. A focus on biologically relevant effect sizes was already set one step before, by filtering for miRNAs with logfold changes >=2. We provided now more justification in the manuscript (e.g. methods: pg. 16, lines 517-520).

  1. c) The authors point out that "previous studies on plasma and placental miRNAs have indicated the involvement of miRNAs in the pathophysiology of preeclampsia" (lines 245-247, pages 10-11) and also that "there have been few overlap among these studies and more investigation is required in order to clarify the exact role of preeclampsia-related miRNAs" (lines 249-250, page 11). As multiple miRNAs have been implicated in preeclampsia, it would be useful to further discuss the implications of the body of evidence that has been accumulating on this topic - suggest consultation of Lv et al. 2019 (J Cell Physiol, 234:1052–1061) and Apicella et al. 2019 (Int J Mol Sci, 20:2837). A discussion on the specific role of these miRNAs would add perspective to these observations.

 Response from the Authors: We re-analyzed the literature and read also the suggested publications. Two miRNAs that were differently regulated in preeclampsia in our study have been reported to differ between preeclampsia and healthy controls in other studies before. However, they were detected in placenta or decidua-derived mesenchymal stem cells making a 1:1 comparison difficult. We included additional information and references in the discussion (pg. 12, lines 339-342).

  1. d) In lines 272-273 (page 11), the authors state that only two studies have investigated the functional role of hsa-miR-1270. However, the authors do not include the bibliographic references of such studies. This is essential. Also, I would suggest to check and revise reference citation throughout the text.

 Response from the Authors: We inserted the references in the manuscript (pg. 13, lines 375, 376)  and revised the reference citations.

  1. Methods:
  2. a) In line 383 (page 13), "a detailed description is provided as supplemental material". In the final sentence of the same paragraph, the authors state this in a similar way, thus I would suggest to revise into one single sentence.

 Response from the Authors: We have revised this part as suggested.

  1. b) In line 391 (page 13), the authors mention an alpha level of 0.1 for normality tests. Why did the authors choose this threshold instead of the common threshold of P<0.05, which indeed was used later on moderated t-test? The authors should include an explanation to this.

 Response from the Authors: We thank the author for this advice. We corrected 0.1 to 0.01 and apologize for this spelling mistake (pg. 15, line 498). The test was carried out on the basis of 0.01, which, in line with the argumentation of the reviewer, is more common worldwide (than 0.1). The relatively high stringency selection for this threshold was determined in order to exclude those candidates from the further analysis who have a highly significant no normal distribution and therefore could possibly be artificial.

  1. c) The sentence in lines 399-400 (page 14) should be revised in regard to phrasing.

 Response from the Authors: We shortened and rephrased the sentence (pg. 16, lines 505-506).

  1. d) The authors used an hsa-miR-1270 inhibitor (lines 449, page 15), but do not state which compound/inhibitor. Similarly, the negative control in cell proliferation assays (line 462, page 15) should be defined, was it the vehicle?

 Response from the Authors: We used a small, synthetic, single-stranded RNA molecule designed to specifically bind to and inhibit mature miRNAs from functioning (Thermo Fisher Scientific). The mirVana miRNA Inhibitor let-7c Positive Control was used as a positive control for transfection efficiency when cells were transfected with the miR-1270-inhibitor. Endogenous let-7c miRNA negatively down-regulate HMGA2 mRNA in cultured cells. HMGA2 is a ubiquitously expressed, non-histone, chromatin protein that can modulate gene expression through changes in chromatin architecture. let-7c miRNA down-regulates levels of HMGA2 mRNA. When transfected into human cells, let-7c miRNA inhibitor blocks endogenous let-7c miRNA, resulting in increased levels of HMGA2 mRNA. Thus, let-7c miRNA inhibitor activity can be monitored in human cells using real-time reverse transcription PCR (RT-PCR) to detect HMGA2 mRNA (https://www.thermofisher.com/order/genome-database/browse/mimics-inhibitors/keyword/hsa-miR-1270?SID=srch-uc-mimic-hsa-miR-1270&mode=and#filters=). Using TaqMan miRNA assays, you can measure the level of miRNA of interest. If the miRNA is bound to the synthetic miR-inhibitor transfected, the duplex should not be detected, indicating efficient miRNA inactivation. Please note, this only works when the isolated total RNA sample is not heat denatured when RT primer is added, as heat denaturation will result in complex dissociation and subsequent detection of microRNA by TaqMan miRNA assay even when cells have been transfected properly.

mirVana miRNA Inhibitor Negative Control #1 was used as a negative control for experiments using miR-inhibitors as recommended by the company. We transfected this negative control in parallel with our positive control (let-7c Positive Control) and the miR-1270- inhibitor. Target expression from negative control-transfected samples was used as a baseline for evaluation of the effect of the control and experimental miRNA inhibitor on target gene expression. Unfortunately, the negative control sequence is not provided by the company (source: Thermo Fisher Scientific; https://www.thermofisher.com/de/de/home.html). We have now included a short explanation in the results part (pg. 10, lines 281-287) of the used materials and samples and included a figure (Supplemental Figure S3).

  1. e) Statistical analysis: which test was used to assess the normality of distribution, Kolmogorov-Smirnov (lines 465-466, page 15), Shapiro-Wilk (lines 389-390, page 13) or both? This should be clarified.

 Response from the Authors: We thank the author for this comment and corrected the part of statistical analysis. For the test of normality distribution Shapiro-Wilk test was performed (pg. 15, line 496 and pg. 17, line 593).

  1. f) The last sentence of the statistical analysis subsection needs revision, as it begins with a URL.

 Response from the Authors: We revised the sentence (see page 18, line 608).

Additional points:

  1. 73, "preeclampsia (51)" (page 2) - Does (51) correspond to a reference? If so, which reference? (reference list ends with 48 references).

 Response from the Authors: We corrected this mistake and removed “(51)”.

  1. 110, "some 47 miRNAs" (page 4) - Remove "some".

 Response from the Authors: We have removed “some”.

  1. 113, "some 39 miRNAs" (page 4) - Remove "some"

 Response from the Authors: We have removed “some”.

  1. 272, "hitherto" (page 11) - The authors could find a more common word for such meaning, as it is an unusual term in scientific literature.

 Response from the Authors: We have changed the wording to “so far” (pg. 13, line 372).

  1. 402, "passage 3 (P3) and passage 5" (page 14) - Revise into abbreviations, i.e. P3 and P5 (these have been previously defined in the text).

Response from the Authors: We now defined P3 and P5 for the first time in the introduction section (pg. 2, line 78)and revised passage 3 and passage 5 into abbreviations.

  1. 414, "fold change (FC)" (page 14) - This abbreviation is not necessary (only appears in schemes inside Figures 1A, 1B and 1C); also it could create confusion with ECFC.

Response from the Authors: We have removed the abbreviation in the text as suggested.

  1. 479-482 (page 15) - Check text formatting.

 Response from the Authors: We checked the formatting and revised the font which was not consistent. The formatting of the manuscript has been performed by the MDPI staff after the submission so we are not sure if the choice of different fonts (Palatino vs. Palatino Linotype) was a conscious decision of the MDPI staff or by accident.

In Supplemental Table S3A, the header states "Sequenz", please revise it.

 Response from the Authors: We revised the word.

References:

  1. Brennan, G.P.; Vitsios, D.M.; Casey, S.; Looney, A.M.; Hallberg, B.; Henshall, D.C.; Boylan, G.B.; Murray, D.M.; Mooney, C. RNA-sequencing analysis of umbilical cord plasma microRNAs from healthy newborns. PLoS One 2018, 13, e0207952, doi:10.1371/journal.pone.0207952.
  2. Khoo, C.P.; Roubelakis, M.G.; Schrader, J.B.; Tsaknakis, G.; Konietzny, R.; Kessler, B.; Harris, A.L.; Watt, S.M. miR-193a-3p interaction with HMGB1 downregulates human endothelial cell proliferation and migration. Scientific reports 2017, 7, 44137, doi:10.1038/srep44137.
  3. Enquobahrie, D.A.; Abetew, D.F.; Sorensen, T.K.; Willoughby, D.; Chidambaram, K.; Williams, M.A. Placental microRNA expression in pregnancies complicated by preeclampsia. Am J Obstet Gynecol 2011, 204, 178.e112-121, doi:10.1016/j.ajog.2010.09.004.
  4. Zhao, G.; Zhou, X.; Chen, S.; Miao, H.; Fan, H.; Wang, Z.; Hu, Y.; Hou, Y. Differential expression of microRNAs in decidua-derived mesenchymal stem cells from patients with pre-eclampsia. J Biomed Sci 2014, 21, 81, doi:10.1186/s12929-014-0081-3.

Round 2

Reviewer 2 Report

Thank you for your detailed point-to-point response and for addressing my comments. I just have one last question regarding the number of healthy pregnancies used in the study. According to section 4.1 in the materials and methods, page 15, line 453-454, '9 healthy pregnancies (n = 6 from maternal; n = 6 from cord blood) were isolated'. So would this mean 12 healthy pregnancies instead of 9? Table 1 also indicates n = 6 from maternal; n = 6 from cord blood.

Author Response

We thank the reviewer for the additional comment and have addressed the question accordingly.

Thank you for your detailed point-to-point response and for addressing my comments. I just have one last question regarding the number of healthy pregnancies used in the study. According to section 4.1 in the materials and methods, page 15, line 453-454, '9 healthy pregnancies (n = 6 from maternal; n = 6 from cord blood) were isolated'. So would this mean 12 healthy pregnancies instead of 9? Table 1 also indicates n = 6 from maternal; n = 6 from cord blood.

Response: We revised the explanation in the text and would like to provide further explanation. We recruited 9 women with healthy pregnancies. From three healthy participants we were able to isolate ECFC from maternal blood as well as from cord blood (3 maternal-cord blood pairs). For the other 6 participants we were able to isolate either maternal (n = 3) or cord blood ECFC (n = 3). Therefore, for the final groups we had ECFC of n = 6 from maternal and of n = 6 from cord blood available.

Reviewer 3 Report

The authors have addressed the majority of the issues that were raised. However, a few points still require further discussion and explanation:

  1. Regarding the point on the filtering thresholds that were used to screen for miRNAs, the authors added the fold change greater than 2 as a filtering threshold to the most significant, i.e. lowest P-value (page 16, lines 517-519). Based on this, the authors identified and chose the following for quantitative RT-PCR validation: hsa-miR-1270 (FC=-2.63, P=0.008) and hsa-miR-2467-5p (FC=3.19, P=0.005) in cord blood; hsa-miR-214-5p (FC=-10.66, P=0.002) and hsa-miR-3177-5p (FC=2.19, P=0.001) in maternal blood. However, there are other miRNAs that presented a higher P-value for the fold change: hsa-miR-3911 in cord blood (FC=-2.34, P=0.007); hsa-miR-214-3p (FC=-8.89, P=0.005), hsa-miR-199a-3p (FC=-8.52, P=0.005), hsa-miR-199b-3p (FC=-8.52, P=0.005), hsa-miR-139-3p (FC=-8.25, P=0.009), hsa-miR-4728-3p (FC=5.06, P=0.003), hsa-miR-4511 (FC=3.45, P=0.007) and hsa-miR-3128 (FC=3.18, P=0.002) in maternal blood. What was the rationale for discarding all these miRNAs and only consider the previously mentioned?
  2. Also considering this point, the authors must not state "we focused on hsa-miR-1270 as it was the most different miRNA (...)" (page 12, lines 328-330). Although a justification was made for the use of P-value, P-value does not provide a measure of difference (given by the fold change) but instead a statistical measure of the significance of such difference. Looking at Table 2, a higher difference was observed for hsa-miR-4726-5p (FC=-2.93 vs FC=-2.63 for hsa-miR-1270), for example. Therefore, the authors must revise the sentence. Also, consider the rationale provided in response to the question above.

Minor point:

Supplemental Table 3A: the header still states "Sequenz", even though the authors stated that it has been revised.

Author Response

We thank the reviewer for additional questions and have addressed these accordingly.

 The authors have addressed the majority of the issues that were raised. However, a few points still require further discussion and explanation:

  1. Regarding the point on the filtering thresholds that were used to screen for miRNAs, the authors added the fold change greater than 2 as a filtering threshold to the most significant, i.e. lowest P-value (page 16, lines 517-519). Based on this, the authors identified and chose the following for quantitative RT-PCR validation: hsa-miR-1270 (FC=-2.63, P=0.008) and hsa-miR-2467-5p (FC=3.19, P=0.005) in cord blood; hsa-miR-214-5p (FC=-10.66, P=0.002) and hsa-miR-3177-5p (FC=2.19, P=0.001) in maternal blood. However, there are other miRNAs that presented a higher P-value for the fold change: hsa-miR-3911 in cord blood (FC=-2.34, P=0.007); hsa-miR-214-3p (FC=-8.89, P=0.005), hsa-miR-199a-3p (FC=-8.52, P=0.005), hsa-miR-199b-3p (FC=-8.52, P=0.005), hsa-miR-139-3p (FC=-8.25, P=0.009), hsa-miR-4728-3p (FC=5.06, P=0.003), hsa-miR-4511 (FC=3.45, P=0.007) and hsa-miR-3128 (FC=3.18, P=0.002) in maternal blood. What was the rationale for discarding all these miRNAs and only consider the previously mentioned?

Response: We thank the reviewer for pointing us to this again. Since it was not feasible to examine all significantly changed miRNAs within this project in detail, we chose the following procedure. First, a focus on biologically relevant effect sizes was set by filtering for miRNAs with logfold changes ³2 independent of the highest effect size (highest fold change). Second, in the groups with logfold changes ³2 we followed the most significant hit (lowest P value), independent of its effect size, which is a common procedure in case-control association studies. Third, the highest fold change for further selection was used (pg. 16, lines 517-522)

  1. Also considering this point, the authors must not state "we focused on hsa-miR-1270 as it was the most different miRNA (...)" (page 12, lines 328-330). Although a justification was made for the use of P-value, P-value does not provide a measure of difference (given by the fold change) but instead a statistical measure of the significance of such difference. Looking at Table 2, a higher difference was observed for hsa-miR-4726-5p (FC=-2.93 vs FC=-2.63 for hsa-miR-1270), for example. Therefore, the authors must revise the sentence. Also, consider the rationale provided in response to the question above.

Response: We changed the sentence to “ …we focused on hsa-miR-1270 as it was one of the most statistically different miRNAs with high fold change (pg. 12, line 329).

Minor point:

Supplemental Table 3A: the header still states "Sequenz", even though the authors stated that it has been revised.

Response: We apologize for this oversight and have revised “sequenz” to “sequence”.
